# Development of a versatile high-throughput mutagenesis assay with multiplexed short-read NGS using DNA-barcoded *supF* shuttle vector library amplified in *E. coli*

**Hidehiko Kawai[1,2]\*, Ren Iwata[2], Shungo Ebi[2], Ryusei Sugihara[2], Shogo Masuda[1], Chiho Fujiwara[2], Shingo Kimura[3], Hiroyuki Kamiya[1,2]\***

[1]Graduate School of Biomedical and Health Sciences, Hiroshima University, Hiroshima, Japan; [2]School of Pharmaceutical Sciences, Hiroshima University, Hiroshima, Japan; [3]Analysis Center of Life Science, Natural Science Center for Basic Research and Development, Hiroshima University, Hiroshima, Japan

**Abstract** A forward mutagenesis assay using the *supF* gene has been widely employed for the last several decades in studies addressing mutation frequencies and mutation spectra associated with various intrinsic and environmental mutagens. In this study, by using a *supF* shuttle vector and non-SOS-induced *Escherichia coli* with short-read next-generation sequencing (NGS) technology, we present an advanced method for the study of mutations, which is simple, versatile, and cost-effective. We demonstrate the performance of our newly developed assay via pilot experiments with ultraviolet (UV) irradiation, the results from which emerge more relevant than expected. The NGS data obtained from samples of the indicator *E. coli* grown on titer plates provides mutation frequency and spectrum data, and uncovers obscure mutations that cannot be detected by a conventional *supF* assay. Furthermore, a very small amount of NGS data from selection plates reveals the almost full spectrum of mutations in each specimen and offers us a novel insight into the mechanisms of mutagenesis, despite them being considered already well known. We believe that the method presented here will contribute to future opportunities for research on mutagenesis, DNA repair, and cancer.

**\*For correspondence:**
kawaih@hiroshima-u.ac.jp (HK);
hirokam@hiroshima-u.ac.jp (HK)

**Competing interest:** The authors declare that no competing interests exist.

## Editor's evaluation

This is a comprehensive description of a technical advance for the analysis of mutations based on the most widely used system for reporting mutations in mammalian, including human, cells. As costs for NGS decline it is likely to become the approach of choice.

## Introduction

In the last half century, various methods such as the Ames test and the micronucleus test have been employed as simple, sensitive, and reliable approaches for evaluation of the mutagenicity of chemicals and drugs (*OECD, 2016*; *OECD, 2020*). More recently, the technological advancement in science has allowed us, in addition to evaluation of mutagenicity, to address the distinct spectra of mutations produced by a broad range of intrinsic and environmental mutagens or anticancer drugs (*David, 2020*). The mutational signatures extracted from these mutation spectra have provided valuable

information about the role of endogenous and exogenous risk factors in the development of many human disorders and have been crucial in finding potential targets for treatment (*Nik-Zainal et al., 2012*; *Meier and Gartner, 2014*; *Koh et al., 2021*).

Until the present day, a vast amount of diverse studies focused on the mechanisms of mutagenesis and DNA repair from the chemical, biochemical, and biological aspects have heavily relied on experimental strategies using cloned genes (*Razzaque et al., 1983*; *Drinkwater and Klinedinst, 1986*; *Gossen et al., 1989*; *White et al., 2019*). One such widely used system is the SV40-based shuttle vector mutagenesis assay, which utilizes vector plasmids containing the *supF* gene of *Escherichia coli* and an indicator bacteria strain (*Parris and Seidman, 1992*; *Parris et al., 1994*; *Sarkar et al., 1984*; *Yoon et al., 2019*; *Suzuki et al., 2022*). The *supF* gene codes for an amber suppressor transfer RNA (tRNA) that translates the amber stop codon (UAG) to tyrosine (*Camakaris and Pittard, 1973*). Depending upon the position and type of mutations in the *supF* gene, the *supF* tRNA cannot translate selectable marker genes with an amber mutation in its indicator *E. coli* strains, such as KS40/pOF105 (*Normanly et al., 1986*; *Obata et al., 1998*). Because of its properties, the *supF* gene is particularly useful for acquiring mutation spectra and identifying mutation signatures, and provides reliable data on mutations generated by specific DNA lesions on the leading or the lagging template DNA strand via translesion synthesis (TLS) (*Drinkwater and Klinedinst, 1986*). However, one needs to be aware of the limitation of the phenotypic mutation spectrum of the *supF* gene; in our experiments for example, from the entire base sequence of the *supF* gene only about 30% can be detected for dual-antibiotic selection of indicator *E. coli*, that is, whether these sites can be identified or not is largely dependent on the experimental conditions, including the scale of the experiment, reporter genes for the selection, the mutagens used, the frequency of mutations, and the efficiency of the detection (*Obata et al., 1998*; *Kraemer and Seidman, 1989*; *Fukushima et al., 2020*). Recently, we reported the development of a new indicator *E. coli* strain – RF01, that is able to identify mutation spectra more efficiently than other strains used for the *supF* assay, even though it still does not allow the elucidation of trinucleotide mutational signatures because of the limited resolution of the mutation spectrum analysis (*Fukushima et al., 2020*).These limitations can be overcome by advanced techniques, namely next-generation sequencing (NGS), which over the last decade is enabling us to obtain various kinds of information from nucleic acid sequence data, including mutations and three-dimensional organization of the genomic DNA at a single cell level (*Shendure et al., 2017*; *Dutta et al., 2022*; *Arrastia et al., 2022*). On the other hand, the NGS technology also has its limitations, especially for identification and quantification of low-frequency mutations, as it can be difficult to distinguish true mutations from the baseline errors of NGS. In recent years, in order to detect low-frequency mutations, numerous library construction methods and sequencing methods have been developed, such as maximum depth sequencing or duplex sequencing (*Schmitt et al., 2012*; *Jee et al., 2016*; *Sloan et al., 2018*). For every scientific objective, trade-offs between accuracy, cost, and speed of the multiple available approaches must be carefully considered. Importantly, there is a scarcity of methods whose technical simplicity and material availability enables any researchers, not only those specialized in NGS, to accumulate, reproduce, and compare data across research fields. Certainly, the availability of an easy, low-cost assay for accurate analysis of mutational signatures would benefit many laboratories who are deterred by the complexity of the existing methods.

In this study, we attempt to develop a simple mutagenesis assay using a random 12-nucleotide (12-nt) DNA-barcoded *supF* shuttle vector library with non-SOS-induced *E. coli* which can amplify the DNA with 'a high-fidelity replication mechanism' as a template for the NGS sample preparation (*Napolitano et al., 1997*). This promotes cost-benefit, high-throughput, and high accuracy with almost no sequencing errors. In the first part, we present data from the multiplexed NGS assay utilizing the barcoded *supF* shuttle vector library and the BGI-platform that uses the rolling circle replication method, and we compare the results for different conditions of sample preparation. In the later part of this study, we present analysis of UV radiation-induced mutations in direct *E. coli* experiments or after transfection into mammalian cells, as an example of the performance of our newly developed assay. UV radiation is a known environmental carcinogen and causes a characteristic mutational pattern via the formation of pyrimidine-pyrimidine dimers (*Varghese and Wang, 1967*; *Mao et al., 2016*; *Lindberg et al., 2019*). The C to T transition at C<u>C</u>N and T<u>C</u>N trinucleotide sites (the mutated base is underlined) is a mutational signature highly specific to UV-mediated mutagenesis (mutational COSMIC signatures SBS7a-d) (*Alexandrov et al., 2020*). The cytosine in UV-induced

pyrimidine-pyrimidine dimers undergoes deamination to uracil, followed by insertion of an A nucleotide opposite the lesion by error-prone TLS polymerases (*Pfeifer et al., 2005*; *Ikehata and Ono, 2011*; *Roberts and Gordenin, 2014*). The substantial amount of sequence data on UV-induced mutations in mammalian cells obtained in this study allowed us to uncover some insightful information about UV-induced mutagenesis, underscoring the importance of a systematic process for sufficient data collection and analysis. We believe that our new approach using the barcoded *supF* shuttle vector library and NGS technology presents a benefit to future research on mutagenesis, DNA repair, and cancer.

## Results

### Construction of barcoded *supF* shuttle vector libraries for NGS assay

Aiming to develop a more advanced and versatile assay system suitable for a wide range of mutagenesis studies, we attempted to apply a short-read NGS platform with multiplex sample preparation to the conventional *supF* mutagenesis assay. At first, we constructed a series of *supF* shuttle vector plasmids, named pNGS2-K1-4 or -A1-4 (*Figure 1—figure supplement 1*). Each plasmid carries all regions depicted for pNGS-K1 or A1 in *Figure 1A*, including an antibiotic resistance gene for either kanamycin (*km'*) or ampicillin (*amp'*). Different orientations of the *supF* gene and the M13 intergenic region were utilized for the purposes of individual experiments, such as the detection of specific types and sites of DNA damage (*Suzuki et al., 2018*; *Suzuki et al., 2021*; *Suzuki et al., 2022*). To investigate both mutation frequencies and profiles as precisely as possible, *supF* shuttle vector libraries containing random 12-nt ($N_{12}$) barcode sequences ($N_{12}$-BC *supF* library) were generated from pNGS2 to be used for NGS analysis. An easy and reliable method to construct $N_{12}$-BC *supF* libraries with sufficient complexity by using single-stranded DNA was developed in this study, as is described in detail in the Materials and methods section. To prepare the samples for NGS analysis, a region from the $N_{12}$-BC *supF* library (*Figure 1B*) was amplified by PCR by using the indicated primer sets (*Figure 1—source data 1*). We opted to use PCR amplification instead of acoustic shearing by ultrasound, in order to avoid inducing oxidative damage to the DNA (*Foox et al., 2021*; *Pugh et al., 2013*; *Ma et al., 2019*). Also, each primer pair for PCR amplification contains in one of the primers a pre-designed 6-nt index sequence ($N_6$) for multiplexed NGS assay, which is used for classification of independent samples in every NGS run. The independently amplified PCR products for each sample were purified and combined in the proper proportion, after which they were subjected to multiplexed deep sequencing in 200 bp paired-end mode on the NGS using a BGISEQ platform (*Foox et al., 2021*). We have also carried out several pilot experiments using 150 bp paired-end mode and/or Illumina platform, which confirmed that these modalities can also be successfully employed for the assay. The procedure for the *supF* mutagenesis NGS assay in this study is illustrated in *Figure 1C*. Depending on the purpose and the design of each experiment, the $N_{12}$-BC *supF* library or cells can be treated with mutagens, and any or all of the libraries (i)–(iv) in *Figure 1C* can be subjected to NGS analysis.

### Analysis for potential bias during NGS sample preparation

To address the issue of detection bias, where the number of $N_{12}$ barcode ($N_{12}$-BC) sequences or variant sequences detected by NGS may be affected by the number of $N_{12}$-BC sequences per library in different experimental conditions, $N_{12}$-BC *supF* libraries with different numbers of $N_{12}$-BC sequences (approximately $1 \times 10^2$ in Lib$10^2$_#1; $1 \times 10^2$ in Lib$10^2$_#2; $1 \times 10^3$ in Lib$10^3$, and $1 \times 10^4$ in Lib$10^4$) were prepared from pNGS2-K1 and analyzed. The background *supF* mutant frequency of each library was first checked by the conventional *supF* assay, and was confirmed to be ~$10^{-5}$, indicating that there was no *supF* phenotypic mutation in the libraries. After that, different amounts of PCR template (1.0, 3.3, 10, 33, or 100 ng of the $N_{12}$-BC *supF* library) were used for PCR amplification and analyzed by NGS, as described in the Materials and methods section. The PCR products were purified and mixed in the optimal ratio so that NGS analysis yields approximately 200–250 reads per barcode sequence. The $N_{12}$-BC sequences for each sample were error-corrected based on a whitelist (*Smith et al., 2017*), and classified by the identical $N_{12}$-BC sequence. The number of unique $N_{12}$-BC sequences in each library is shown in *Figure 2A* (left-side graph in each pair of bar plots). The number of identified unique $N_{12}$-BC sequences was 117–119 for Lib$10^2$_#1, 97 for Lib$10^2$_#2, 1421–1426 for Lib$10^3$, and 13125–13172 for Lib$10^4$, very close to the estimation based on the number of colonies used to prepare each library.

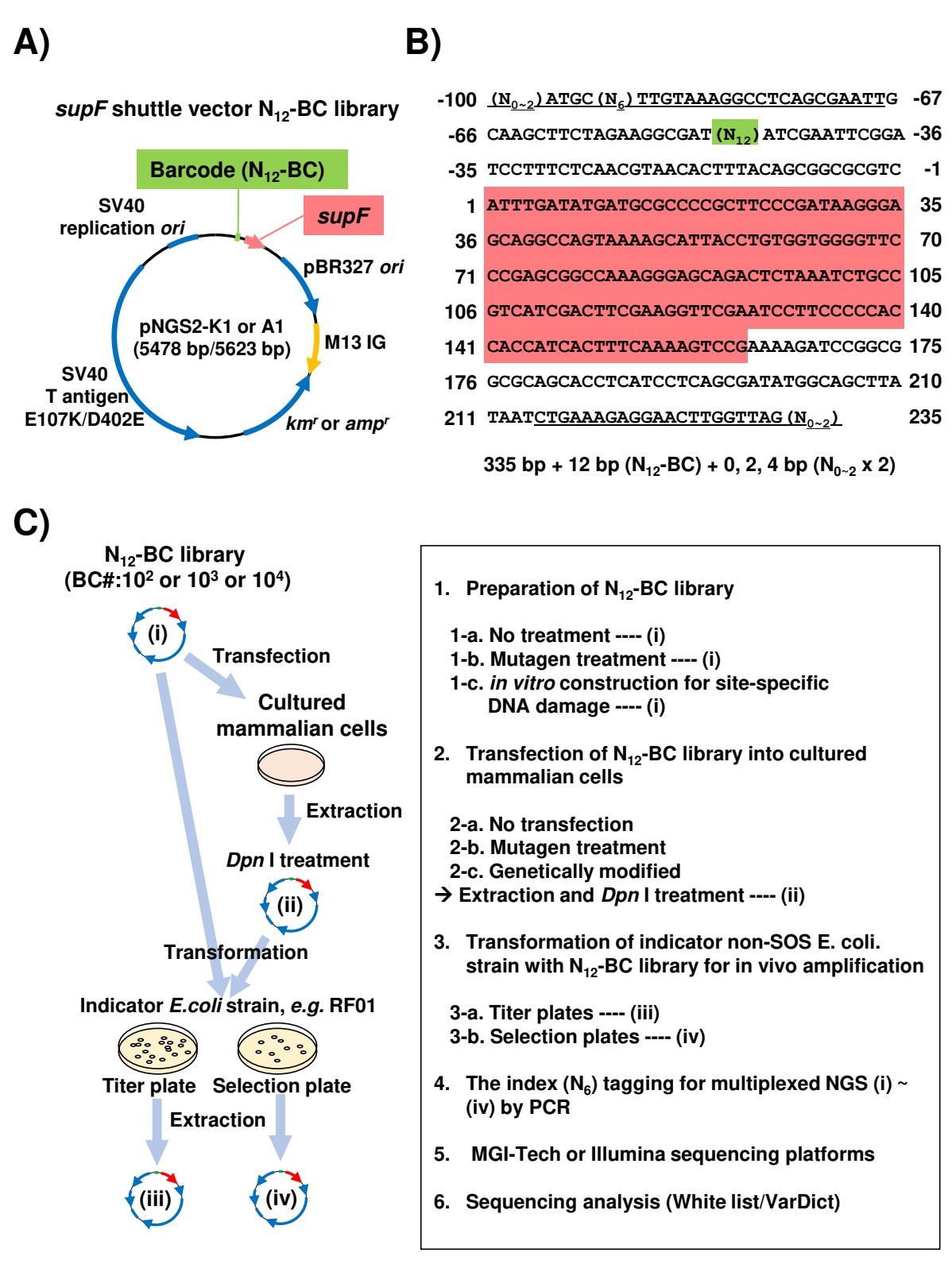

**Figure 1.** Strategy for the *supF* mutagenicity next-generation sequencing (NGS) assay. (**A**) Schematic map of the shuttle vector pNGS2-K1 or -A1. The shuttle vector plasmid DNA encodes the amber suppressor transfer RNA (tRNA) (supF) gene, the TP53-/Rb-binding-deficient mutant SV40 large T antigen (E107K/D402E) gene, the SV40 replication origin, the pBR327 origin of replication, the M13 intergenic region, the kanamycin or ampicillin resistance gene (*km^r* and *amp^r*), and the randomized 12-nucleotide barcode DNA sequence (N_{12}–BC). (**B**) Nucleotide sequence of the non-transcribed

*Figure 1 continued on next page*

*Figure 1 continued*

strand (sense strand) of the DNA fragment from $N_{12}$-BC libraries amplified by PCR in preparation for NGS. The underlined letters indicate the primer set for PCR amplification (*Figure 1—source data 1*). All primer sets contain 0, 1, or 2 random nucleotides at the 5' end to ensure proper signal detection for amplicon sequencing with a short-read NGS platform. A pre-designed 6-nucleotide sequence ($N_6$) in one of the primers serves as an index sequence for distinguishing multiplex samples in a single sequencing run. The $N_{12}$ (green background color) represents the $N_{12}$-BC DNA sequence used for deep-sequencing data analysis to identify variants from identical template shuttle vectors. The sequence with red background indicates the *supF* gene starting at position 1. (**C**) Schematic illustration of the mutagenicity NGS assay procedure. One or more of the $N_{12}$-BC libraries (i)–(iv) can be applied for the sequencing analysis.

The online version of this article includes the following source data and figure supplement(s) for figure 1:

**Source data 1.** The nucleotide sequences of primer sets for multiplexing sample preparation for next-generation sequencing (NGS).

**Figure supplement 1.** SupF shuttle vector $N_{12}$-BC library.

The number of reads was also within the expected range, and comparable at different amounts of PCR template (*Figure 2A*, right-side graph in each pair of bar plots). This data suggests that most $N_{12}$-BC sequences could be detected and classified, and that there was almost no overlapping of $N_{12}$-BC sequences in *E. coli* colonies used for library preparation. The actual detected number of reads for each identical $N_{12}$-BC sequence was compared in stacked bar graphs in *Figure 2B*. There was no correlation between the amount of PCR template and the number of reads for each identical $N_{12}$-BC sequence, confirming that an appropriate PCR procedure can evenly amplify the reads from the plasmid DNA template in the libraries, independently from the template amount.

## Variant calls from aligning reads to the reference sequence

The variant sequences to a reference sequence were detected by VarDict, and the threshold for allele frequency was set to 0.1 (*Lai et al., 2016*). In this study, the allele frequency for each classified $N_{12}$-BC sequence was referred to as variant frequency (VF). The VF has to be considered and interpreted depending on the experimental conditions and purposes. The read counts for individual $N_{12}$-BC sequences with variants were graphed on a scatter plot as a function of VFs (*Figure 2—figure supplement 1A*). The VFs appeared to be distributed in a bipolar manner, but no correlation was observed between the number of reads and VFs, suggesting that a low number of reads cannot be the cause for lower VFs. For all samples, about 20–40% of $N_{12}$-BC sequences contained reads with variants (VFs equal to or greater than 0.1). In *Figure 2—figure supplement 1B*, the distribution of the number of $N_{12}$-BC sequences with a variant by VF is presented as bar histograms. Based on this data we decided in the experiments hereafter to divide the $N_{12}$-BC sequences with variants into two subgroups along the line of 0.4 of the VF.

## VFs and baseline sequencing errors

In *Figure 3A–C*, applying a threshold VF of 0.4, the number of $N_{12}$-BC sequences with variants above and below the threshold were plotted in separate graphs (*Figure 3A and C*, respectively) with respect to their position in the sequencing region shown in *Figure 1B*. It becomes obvious that almost all $N_{12}$-BC sequences with variant reads and VFs 0.4–1.0 were located in the region between –80 and –20. This region is the cloning region for the random 12-nt barcoding and does not manifest as *supF* mutant phenotype by indicator *E. coli* in the conventional *supF* assay. These mutations most likely originate from the oligonucleotides used for the $N_{12}$-BC sequence insertion during preparation of the $N_{12}$-BC *supF* library. The types of variant sequences in this region are indicated by different colors in *Figure 4B*. Notably, the proportions of variant calls are practically identical at different template amounts for the same $N_{12}$-BC *supF* library, which means that the pattern of variant types can provide a fingerprint property for each $N_{12}$-BC *supF* library and a practical index for the detection of mutations.

On the other hand, the $N_{12}$-BC sequences with variant reads in the range of VF 0.1–0.4 were almost exclusively located at either or both positions 55 (a single nucleotide substitution [SNS] from A to C) and 69 (from T to C) of the *supF* gene, as shown in *Figure 3C and D*. Also, almost all $N_{12}$-BC sequences with variants at these positions had VF < 0.4 (*Figure 3D*). Therefore, variant sequences at these positions were assumed to be sequencing errors and not true mutations. The percentage of reads with 0.1–0.4 VFs for each position is shown in *Figure 4E* (the value is the median from all 20 libraries). Positions 55 and 69 contained variant calls in 24.7% and 42.6% of reads, respectively. Taken together, this data suggests that as long as the number of harvested colonies is less than the number

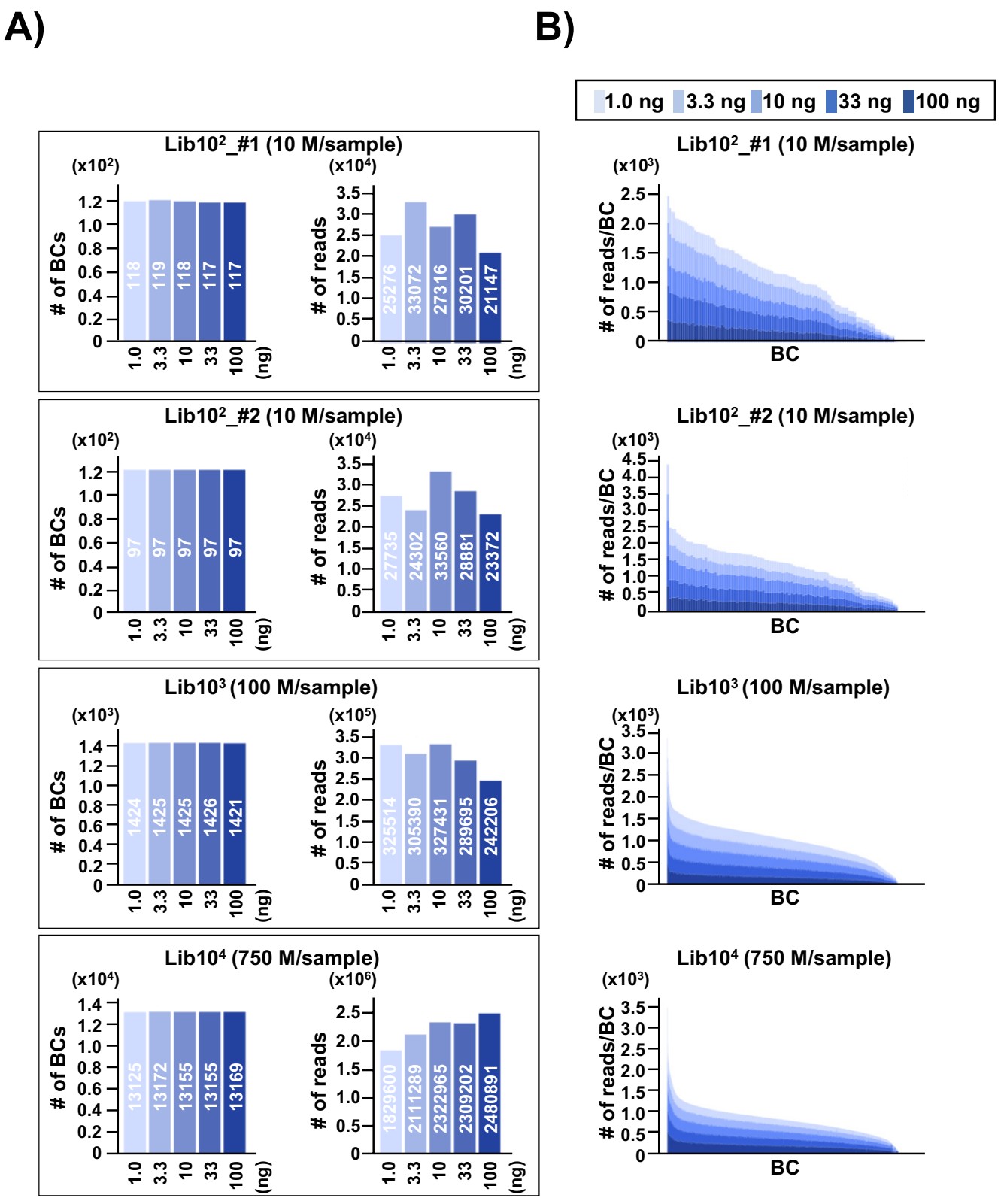

**Figure 2.** Number of classified unique $N_{12}$-BC sequences in the multiplexed next-generation sequencing (NGS) libraries prepared by PCR. (**A**) In the caption of each pair of graphs, the '$10^n$' ($Lib10^2$_#1, $Lib10^2$_#2, $Lib10^3$, and $Lib10^4$) indicates the approximate number of colonies used for the $N_{12}$-BC library preparation from pNGS2-K1, and the '$n$ M/sample' indicates expected '$n$' million reads per individual sample in the multiplexed NGS. The left-side graph in each of the four pairs represents the number of classified $N_{12}$-BC sequences in indexed samples prepared by PCR with different amounts

*Figure 2 continued on next page*

Figure 2 continued

of template (1.0, 3.3, 10, 33, and 100 ng, indicated by graded blue colors). The right-side graph shows the observed number of reads for each indexed sample. Source data is available in **Supplementary file 1A-1D**. (**B**) Coverage distributions for each unique $N_{12}$-BC sequence for the samples in (**A**).

The online version of this article includes the following figure supplement(s) for figure 2:

**Figure supplement 1.** Variant frequency (VF) in $N_{12}$-BC sequences from the multiplexed next-generation sequencing (NGS) libraries.

of identified $N_{12}$-BC sequences, VF around 0.4 can be used as the threshold value between baseline sequencing errors and true mutations.

In addition, almost identical results to those shown in **Figures 2 and 3** were obtained from a series of $N_{12}$-BC libraries originating from pNGS2-A1 containing ampicillin resistance instead of the kanamycin resistance gene (raw data is available in DDBJ database).

## Performance of the NGS mutagenesis assay using $N_{12}$-BC *supF* libraries

Next, two independently prepared $N_{12}$-BC $10^4$ libraries from pNGS-K3, Lib10$^4$_#1 (#1) and Lib10$^4$_#2 (#2), were used to evaluate the performance of the newly developed mutagenesis assay. To produce different numbers of characteristic DNA lesions, the libraries were irradiated with UV-C at 0, 50, 100, 200 J/m$^2$, and then directly transformed into the indicator *E. coli* strain RF01 which is derived from *E. coli* KS40 under non-SOS-induced conditions (**Napolitano et al., 1997**). The RF01 transformed with irradiated or non-irradiated libraries were diluted and plated on LB plates with or without antibiotics (selection or titer plates, respectively), as for a conventional *supF* assay. As expected, the number of RF01 colonies on the titer plates was obviously decreased by UV irradiation in a dose-dependent manner (the average number of colonies per ng of the $N_{12}$-BC $10^4$ library, from three experiments, were $5.0 \times 10^6$ for 0 J/m$^2$, $3.6 \times 10^6$ for 50 J/m$^2$, $1.7 \times 10^6$ for 100 J/m$^2$, and $3.5 \times 10^4$ for 200 J/m$^2$), and on the selection plates containing nalidixic acid and streptomycin the mutant frequencies were in the order of $10^{-4}$ to $10^{-5}$. Approximately 10,000 colonies of RF01 transformed with $N_{12}$-BC *supF* libraries were harvested from one to three titer plates, libraries were extracted, and approximately 400 Mb per sample were analyzed by NGS. The number of $N_{12}$-BC sequences shown in **Figure 4A** were identified as unique $N_{12}$-BC sequences, indicating that even though some $N_{12}$-BC sequences were overlapping between colonies, the majority were not. The number of $N_{12}$-BC sequences with a variant and VF 0.4–1.0, as well as their distribution in the analyzed region (positions –67 to 214) are shown in **Figure 4B** and **Figure 4—figure supplement 1**, respectively. The mutations seem to be evenly distributed, except for the region around the 12-nt barcoding site. This data suggests that all of the variants in non-irradiated samples were derived from the barcoding process with the custom-synthesized $N_{12}$ oligonucleotides during the $N_{12}$-BC library construction, and the number of $N_{12}$-BC sequences with a variant in the *supF* gene was slightly increased by UV irradiation in a dose-dependent fashion. When the $N_{12}$-BC sequences with a variant were classified according to nucleotide position along the line of position –20, near the start site of the *supF* gene (position –15 is the first nucleotide of the promoter), it is evident that there were no variant calls in the region of the *supF* gene in the non-irradiated library (**Figure 4C and D**). The major type of variant sequences identified by variant calls in the irradiated libraries were SNS (**Figure 4C and E**). Also, more than 80% of the SNSs consisted of G:C to A:T transitions (largest proportion) and G:C to C:G transversions (**Figure 4F**). The mutation frequency for the *supF* gene was calculated as the ratio between the number of variant sequences and the total number of analyzed $N_{12}$-BC sequences (**Figure 4G**). For comparison, the *supF* mutant frequency was also determined by the conventional *supF* assay, as the number of colonies on selection plates versus that on titer plates (**Figure 4H**). The results from the two methods exhibit very similar trends of mutation frequency increasing in a UV-dose-dependent fashion, yet the *supF* mutation frequencies calculated from the NGS data were about 20 times higher than those determined by the conventional *supF* assay. The possible reasons for this discrepancy are the following: (1) a *supF* gene containing a UV-damaged non-transcribed strand and a non-damaged transcribed strand confers a nalidixic acid and streptomycin-sensitive phenotype in individual colonies of RF01 due to the wild-type *supF* tRNA expression; (2) there is a selection bias due to 'silent mutations' or 'phenotypic differences' in the mutant phenotype of the *supF* gene, depending on either nucleotide positions or the substituted bases (for instance, positions 21–60 of the *supF* gene correspond to the pre-tRNA sequence); (3) a plasmid may contain several mutations at different positions.

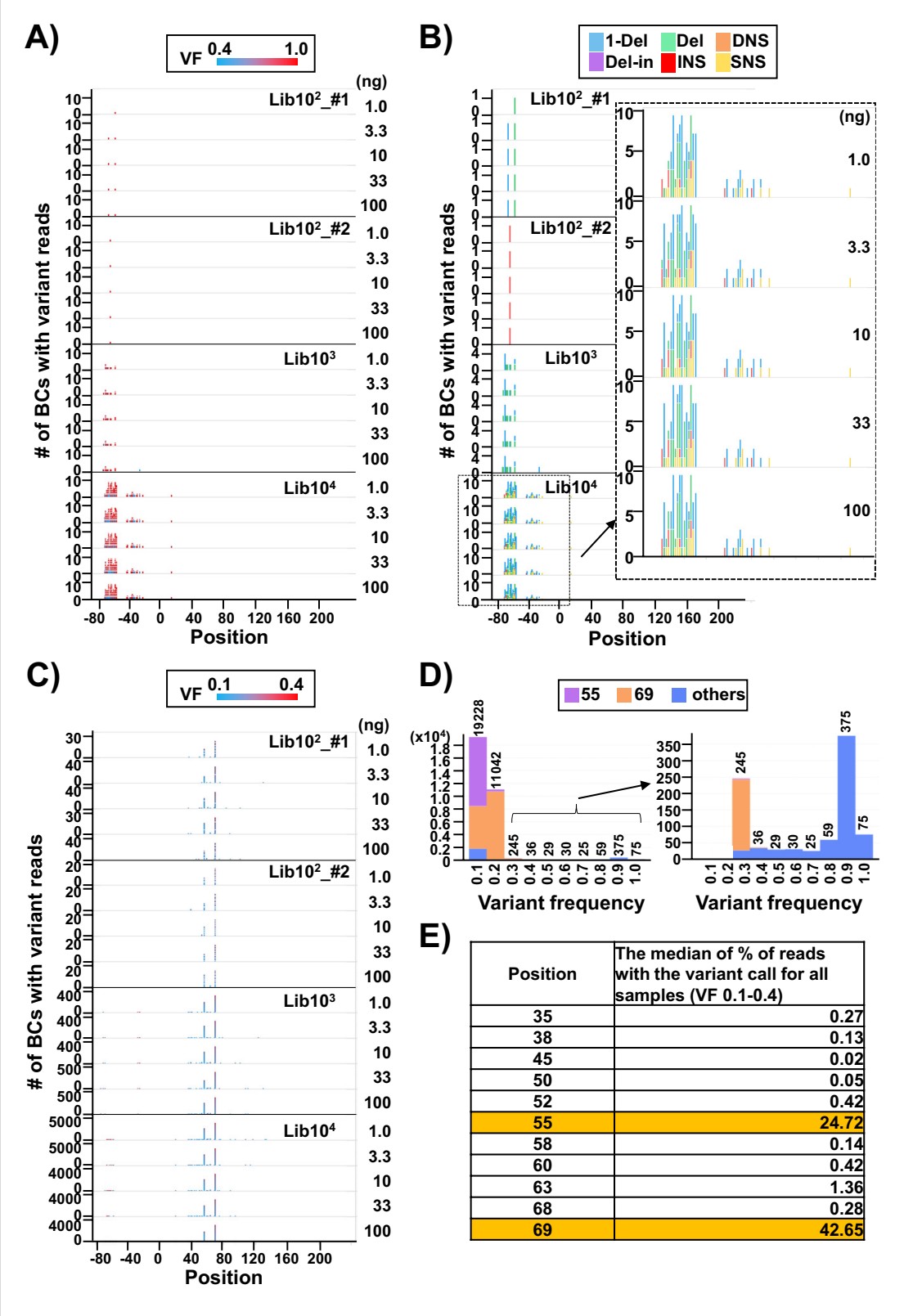

**Figure 3.** Discrimination between true mutations and sequencing errors at the nucleotide positions. (**A**) Number of N$_{12}$-BC sequences with a variant exceeding 0.4 variant frequency (VF) according to their nucleotide positions. The bar colors reflect the VF, as indicated by the heatmap at the top of panel (**A**). Source data is available in ***Supplementary file 2***. (**B**) The same dataset as in panel (**A**), with the bar colors reflecting different variant types. The variant calls are categorized into six types: one-nucleotide deletion (1-Del), deletions larger than one nucleotide (Del), deletions with insertions

*Figure 3 continued on next page*

Figure 3 continued

(Del-in), dinucleotide substitutions (DNSs), insertions (INS), and single nucleotide substitutions (SNSs). The data for the BC10$^4$ library is outlined with a dashed line and enlarged. (**C**) Number of N$_{12}$-BC sequences with a variant in the 0.1–0.4 VF range according to their nucleotide positions. (**D**) Histograms of the distribution of the number of N$_{12}$-BC sequences by VF. The data for the 0.3–1.0 VF range are enlarged on the right side. Different colors indicate the positions of the variant sequence: position 55 (violet), 69 (orange), or other (blue). (**E**) Comparison of the possible sequencing errors at different nucleotide positions in the *supF* gene, presented as percentage of BC sequences with variants below VF 0.4. The value in the right-hand side column of the table is the median from all analyzed samples (n=20).

In order to investigate in more detail the SNS mutation spectrum, all possible 192 trinucleotide contexts of mutations from the titer plates were compared between the complementary DNA strands. The combined data from libraries Lib10$^4$_#1 and Lib10$^4$_#2 is shown in *Figure 5*, top and bottom plot for the non-transcribed and transcribed DNA strand, respectively. The most common of the SNSs was a change at the trinucleotide sequence T<u>C</u>N to either T<u>T</u>N or T<u>G</u>N. In addition, we subjected samples of UV-irradiated N$_{12}$-BC libraries to direct PCR amplification without transformation into *E. coli* and analyzed them. In this case, no dose-dependent increase of the variant sequences was observed (raw data is available in DDBJ database), and the data from all irradiated samples were very similar to the data shown in *Figures 3 and 4*. The G:C to A:T transitions were very rarely detected, indicating that the high fidelity DNA polymerase with no TLS activity used in the PCR process cannot replicate DNA and produce mutations in the presence of lesions generated by UV irradiation, such as cyclobutane pyrimidine dimers (CPDs) or pyrimidine(6–4)pyrimidone photoproducts (6–4PPs).

## Mutation spectra in the UV-irradiated libraries from selection plates

In the case of the conventional *supF* assay, to analyze the sequences of the *supF* gene in the mutants, a number of single colonies on the selection plate were picked up and analyzed by Sanger sequencing. Next, to analyze the sequences of the *supF* gene in the mutants using the newly developed NGS assay, the approximate number of colonies on the selection plate were determined, and the colonies were harvested together. The results are presented in *Figure 6A*. Only a low number of colonies were grown on the selection plates in the non-irradiated condition. The libraries were purified from the collected pellet of *E. coli* and analyzed as described above for the titer plates (16 Mb per sample were analyzed). Again, the number of detected N$_{12}$-BC sequences were almost equal to the number of harvested colonies, and a limited number of N$_{12}$-BC sequences (~2%) contained more than one variant sequence (*Figure 6A*). The positions of the variant sequences are shown in *Figure 6B* and *Figure 6—figure supplement 1*, where the colors indicate the variant types or the VFs, respectively. As expected, in contrast to the titer plates (*Figure 4C*), here the mutation pattern manifested more specific peaks at certain positions, which could be associated with both the *supF* mutation phenotype and the nucleotide signature. Although the mutation/mutant frequency was increased by UV irradiation in a dose-dependent manner (*Figure 4G and H*), the proportion of mutation types (*Figure 6C*) and the sequences of base substitutions (*Figure 6D*) remained very similar at different irradiation doses, as were also the dinucleotide substitutions (DNSs) (*Figure 6E*). The mutation types and sequences were comparable to those found in the samples from titer plates shown in *Figure 4*. However, it is noteworthy that the distributions of the trinucleotide sequences were significantly different between the non-transcribed and transcribed strand of the *supF* gene, contrary to the observations from the titer plates (*Figure 7* versus *Figure 5*). Nevertheless, mutations from both titer and selection plates were almost exclusively C:G to T:A transitions at C<u>C</u>N:N<u>G</u>G to C<u>T</u>N:N<u>C</u>G and T<u>C</u>N:N<u>G</u>A to T<u>T</u>N:N<u>A</u>A.

## UV irradiation mutagenesis in mammalian cells

Finally, to evaluate the performance of the newly developed mutagenesis assay in mammalian cells, the UV-irradiated N$_{12}$-BC 10$^4$ libraries (Lib10$^4$_#1 and Lib10$^4$_#2) were transfected into human osteosarcoma U2OS cells. After 48 hr, the libraries were extracted from the cells, and treated with dam-G$^m$ATC-methylated DNA-specific restriction enzyme *Dpn* I to digest unreplicated DNAs that contain UV photoproducts. The purified libraries were then transfected into the indicator *E. coli* strain RF01, and samples for NGS analysis were prepared from titer and from selection plates, same as described above for the direct RF01 experiments. The data is presented in *Figures 8 and 9* analogously to *Figures 4 and 5*. The trends found from experiments in mammalian cells were comparable to these in *E. coli*, although the data was significantly different in terms of the number of mutations. First, we

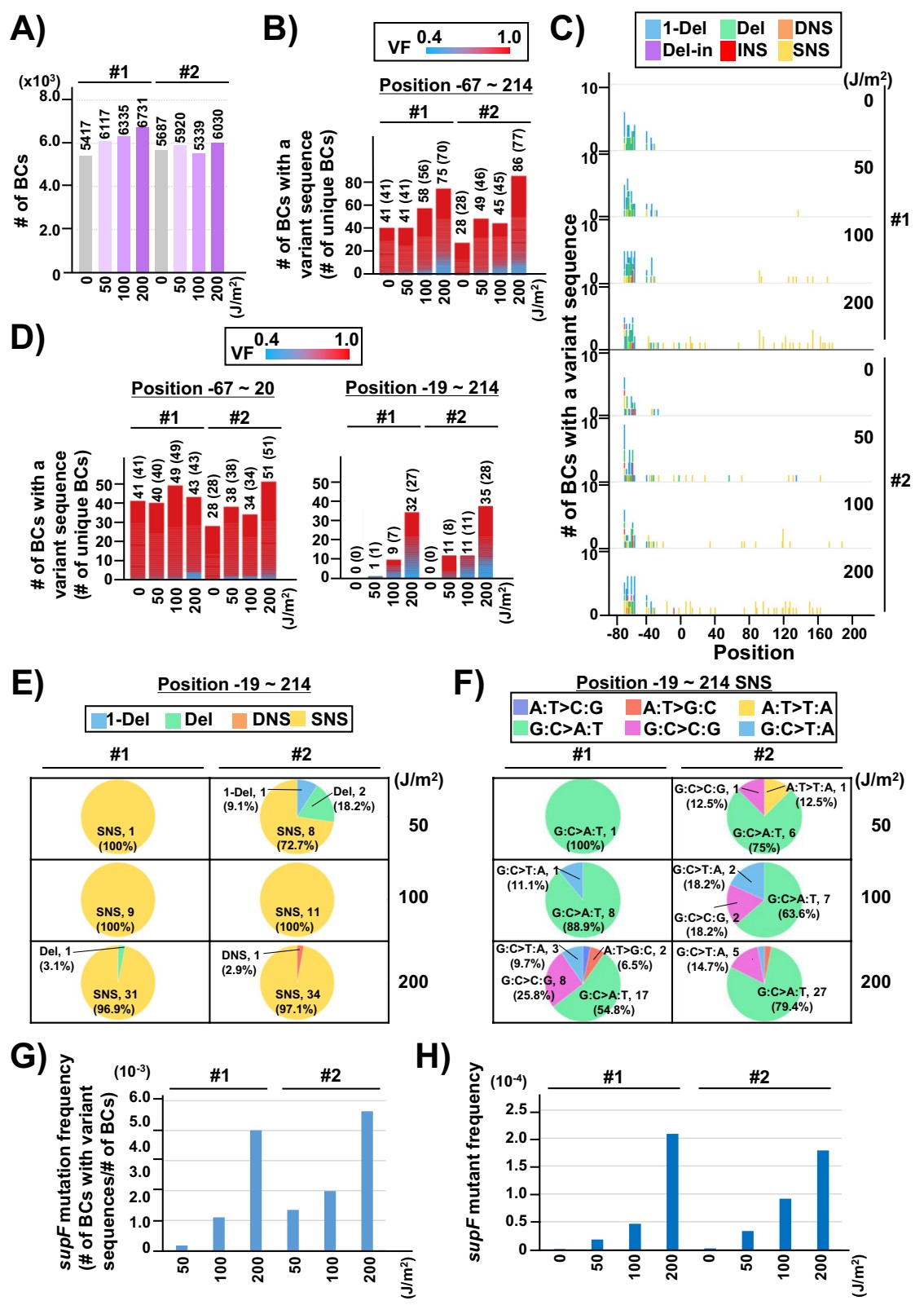

**Figure 4.** Performance of the *supF* mutagenicity next-generation sequencing (NGS) assay in bacterial cells (data from titer plates). (**A**) Number of classified N$_{12}$-BC sequences analyzed from approximately 10$^4$ colonies from titer plates (see *Figure 1C* (iii)). Lib10$^4$_#1 (#1) and Lib10$^4$_#2 (#2) represent two independently prepared BC10$^4$ libraries from pNGS2-K3 irradiated with UV-C at doses of 0, 50, 100, and 200 J/m$^2$ and used for transformation of the indicator *Escherichia coli* strain RF01. Source data is available in *Supplementary file 3A*. (**B**) Number of N$_{12}$-BC sequences with a variant sequence

*Figure 4 continued on next page*

*Figure 4 continued*

exceeding VF 0.4. The number in brackets refers to the number of unique BCs with variant sequences. (**C**) Number of $N_{12}$-BC sequences with a variant exceeding VF 0.4 at different UV-C doses and their distribution by nucleotide position. The variant types are shown in different colors as indicated in the legend. (**D**) The same dataset shown in (**B**) is now separated into two graphs: the graph on the left side represents positions from –67 to –20, and the graph on the right side – positions from –19 to 214 containing the *supF* gene. (**E**) Pie charts of the proportions of mutation types: one-nucleotide deletion (1-Del), deletions larger than one nucleotide (Del), dinucleotide substitutions (DNSs), deletions with insertions (Del-in), insertions (INS), single nucleotide substitution (SNS), categorized as in *Figure 3B*. Source data is available in *Supplementary file 3B*. (**F**) Mutation spectrum shown as pie charts of the proportions of different base substitutions. Source data is available in *Supplementary file 3C*. (**G**) The *supF* mutation frequencies determined by NGS. The mutation frequency is determined by dividing the number of $N_{12}$-BC sequences with a variant sequence at positions from –19 to 214 by the total number of $N_{12}$-BC sequences in the sample (data shown in (**D and A**), respectively). (**H**) The *supF* mutant frequencies determined by the conventional method, that is, by dividing the number of colonies grown on the selection plate by the number of colonies grown on the titer plate.

The online version of this article includes the following figure supplement(s) for figure 4:

**Figure supplement 1.** Number of $N_{12}$-BC sequences with a variant exceeding variant frequency (VF) 0.4 at different UV-C doses and their distribution by nucleotide position (in bacterial cells, data from titer plates).

analyzed the samples from $N_{12}$-BC *supF* libraries transfected into U2OS cells and amplified in the indicator *E. coli* on titer plates. Approximately, from 4000 to 7000 $N_{12}$-BC sequences were identified per sample from around 10,000 colonies transformed with libraries from U2OS cells (*Figure 8A*). A significant dose-dependent increase was observed in the number of $N_{12}$-BC sequences with variant calls, and higher number of variant sequences were obtained from the same number of colonies compared to directly transformed with the libraries *E. coli* (*Figure 8B* versus *Figure 4B*). These differences most likely reflect mutagenic events in U2OS cells arising from UV-induced DNA lesions, mainly from their deamination products. The nucleotide positions and types of variant calls are presented in *Figure 8C* and *Figure 8—figure supplement 1*. It is evident that only a very low number of variant sequences were detected in non-irradiated samples. The detected mutations were distributed throughout the analyzed region, including around the barcoding sites (*Figure 8C and D*). As shown in *Figure 8E*, the major type of variant sequences was SNSs, which is the expected UV-induced mutation signature, and this observation was the same as in directly *E. coli* experiments. However, in the U2OS cells experiment, the percentages of both DNSs and insertions at deletion sites (Del-ins) were increased in a dose-dependent manner. As will be described later, most of the variant sequences detected as Del-in by VarDict were triplet mutations, which are known UV-specific mutations and are defined as multiple base substitutions within a 3-nt sequence containing at least one dipyrimidine. In contrast, the proportions of SNSs were not significantly changed at different irradiation doses and the majority were G:C to A:T transitions (*Figure 8F*). Same as in the direct *E. coli* experiments, the mutation frequency for the *supF* gene was calculated using the ratio between the number of variant sequences and the total number of analyzed $N_{12}$-BC sequences (*Figure 8G*). The *supF* mutant frequencies were also determined by the conventional assay, as the number of colonies on the selection plates versus that on the titer plates (*Figure 8H*). Again, the results exhibit a very similar trend of a dose-dependent increase, and predictably, the *supF* mutation frequencies calculated from NGS data were about 15 times higher than those determined by the conventional assay. As in the direct *E. coli* experiments (*Figure 5*), the SNS mutation spectra were compared between the complementary DNA strands (*Figure 9*). The most common of the SNSs were G:C to A:T transitions, mainly found at TCN:NGA to TTN:NAA.

Next, the samples obtained from $N_{12}$-BC *supF* libraries transfected into U2OS cells and amplified in the indicator *E. coli* on selection plates were analyzed. Contrary to the direct *E. coli* experiments (*Figure 6A*), approximately 1000 colonies were yielded for each sample from selection plates, even in the non-irradiated condition. As shown in *Figure 10A*, in spite of the fact that the samples for all UV doses were prepared from almost the same number of colonies (1000), the identified number of $N_{12}$-BC sequences was less than 1000 and showed an increase in a dose-dependent manner, approaching 1000 only at 200 J/m². This result indicates that the classified number of $N_{12}$-BC sequences with variant sequence(s) were already saturated for the samples irradiated at doses of 0, 50, and 100 J/m². Even if more colonies were obtained from a larger number of plates, the same overlapping $N_{12}$-BC sequences with variant(s) would be identified. In other words, almost all variant sequences were already detected at doses lower than 200 J/m², as long as the indicator *E. coli* transformed with the libraries could form a colony and the sample could be amplified by the PCR procedure. Moreover, the dose dependency of the discrepancy between the numbers of $N_{12}$-BC sequences and the

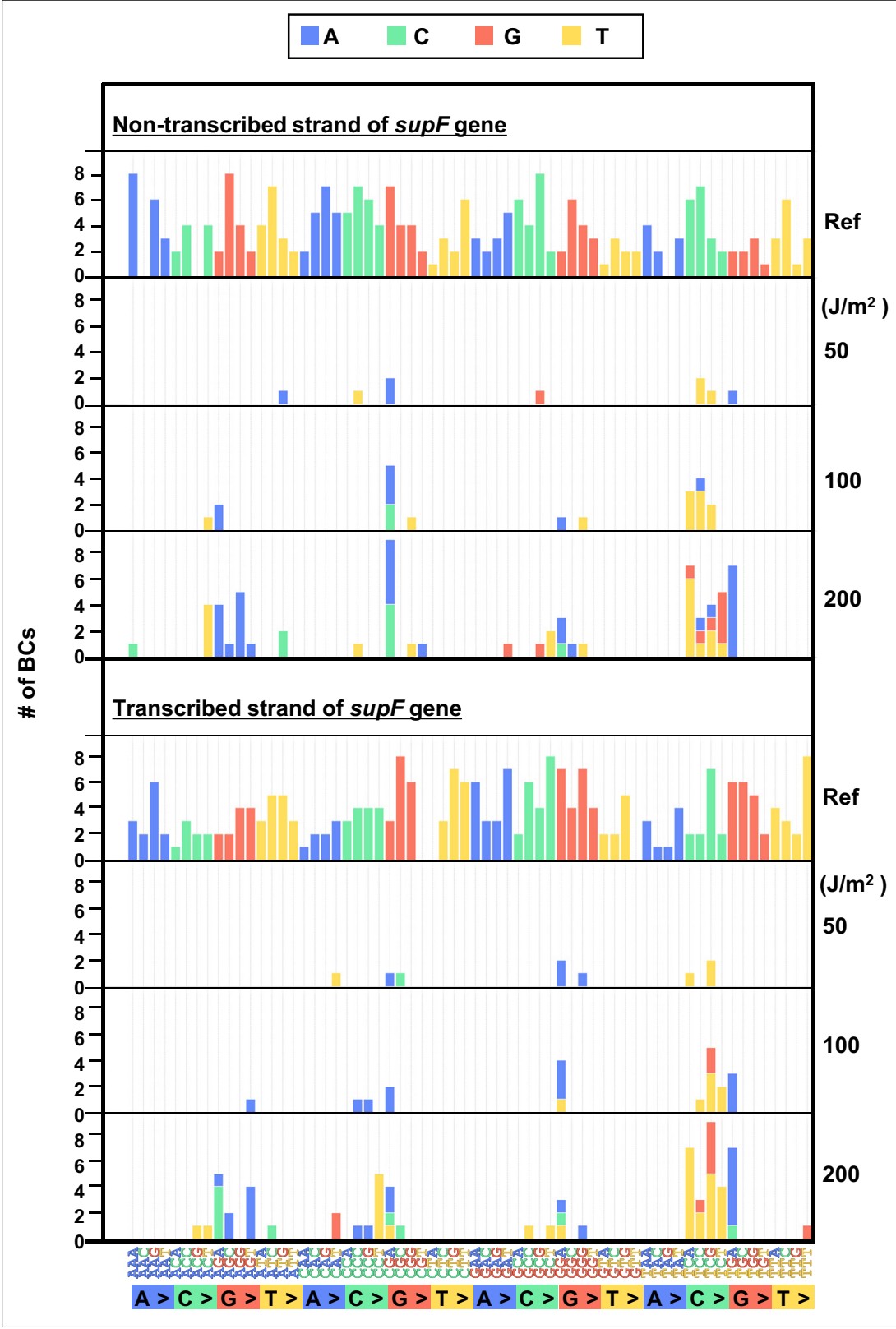

**Figure 5.** The 192 trinucleotide contexts of ultraviolet (UV) radiation-induced single nucleotide substitutions (SNSs) in bacterial cells (data from titer plates). UV radiation-induced SNSs shown in 192 trinucleotide contexts for the non-transcribed (top panel) and transcribed (bottom panel) strand of the *supF* gene (nucleotide positions from –19 to 214). The data is combined from Lib10⁴_#1 and Lib10⁴_#2. The substituted bases are indicated in colors corresponding to the notation on the horizontal axis; the UV doses are shown on the right side of the graph; the reference sequence for the analysis is denoted as Ref Source data is available in *Supplementary file 3D*.

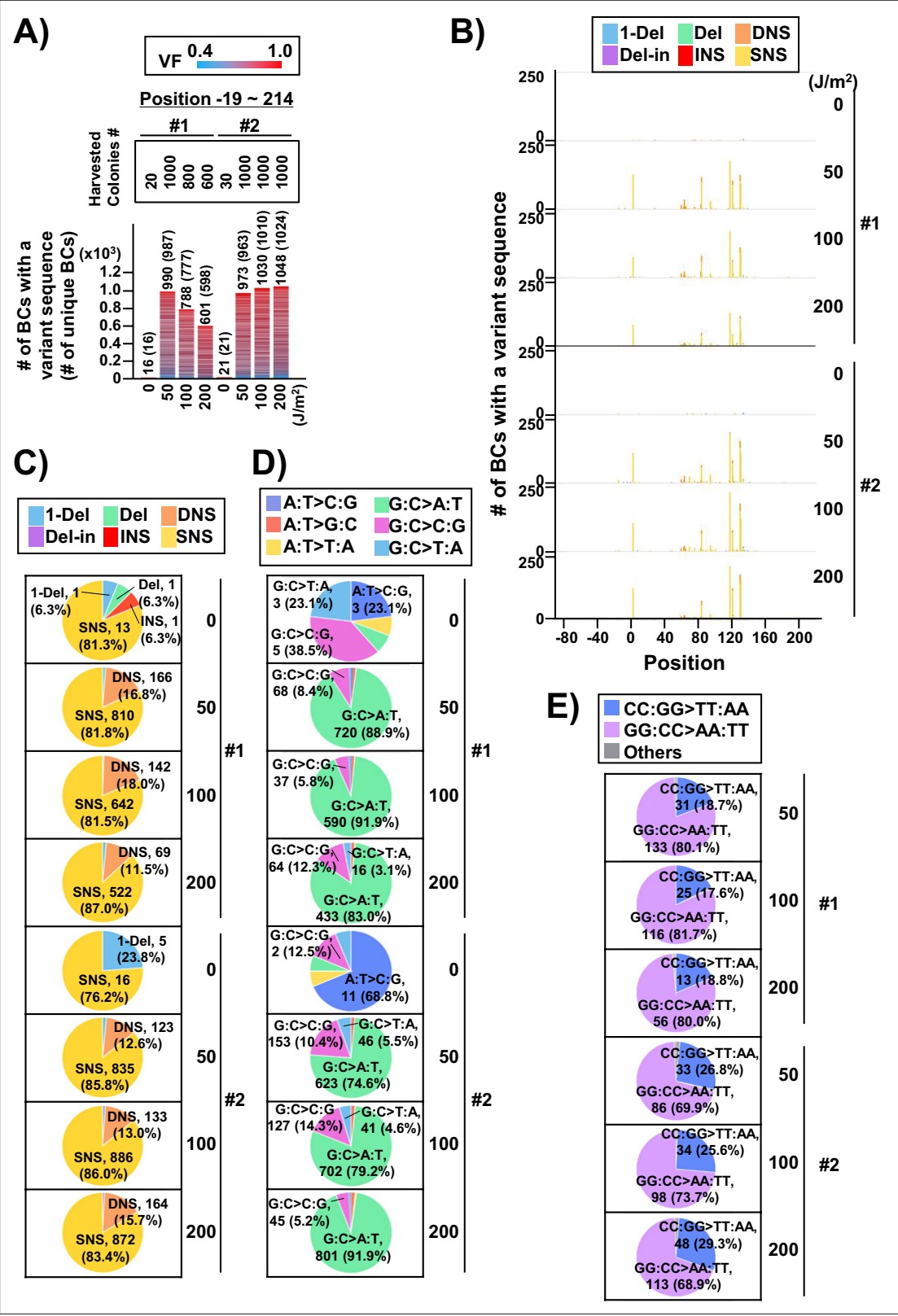

**Figure 6.** Performance of the *supF* mutagenicity next-generation sequencing (NGS) assay in bacteria cells (data from selection plates). (**A**) Number of classified N₁₂-BC sequences from colonies in selection plates (see *Figure 1C* (iv)). The approximate numbers of harvested colonies are shown on top of the graph in a rectangle denoted as 'Harvested colonies #'. The number above each bar represents the number of classified N₁₂-BC sequences, and the number in the brackets refers to the number of unique BCs with variant sequences. Source data is available in *Supplementary file 4A*. (**B**) Number

*Figure 6 continued on next page*

*Figure 6 continued*

of N$_{12}$-BC sequences with a variant exceeding variant frequency (VF) 0.4 at different UV doses and their distribution by nucleotide position. The variant types are shown in different colors as indicated in the legend. (**C**) Pie charts of the percentage distribution of different mutation types. Source data is available in *Supplementary file 4B*. (**D**) Pie charts of the percentage distribution of different single nucleotide substitutions (SNSs). Source data is available in *Supplementary file 4C*. (**E**) Pie charts of the percentage distribution of different dinucleotide substitutions (DNSs). Source data is available in *Supplementary file 4D*.

The online version of this article includes the following figure supplement(s) for figure 6:

**Figure supplement 1.** Number of N$_{12}$-BC sequences with a variant exceeding variant frequency (VF) 0.4 at different UV-C doses and their distribution by nucleotide position (in bacterial cells, data from selection plates).

number of variant sequences indicates that multiple mutations in the same read were produced. The number of mutations per N$_{12}$-BC sequence is shown by a horizontal stacked bar graph in *Figure 10B*, and it can be seen that the proportion of N$_{12}$-BC sequences with multiple mutations was slightly increased in a dose-dependent manner. The positions and types of mutations plotted in *Figure 10C* and *Figure 10—figure supplement 1* reveal that the overall results were very similar to those from *E. coli* directly transformed with N$_{12}$-BC *supF* libraries (*Figure 6B*), the reasons for which are associated with the properties of the *supF* gene and UV-induced DNA lesions. Regarding the proportion of mutation types and SNSs, the data from selection plates (*Figure 10D and E*) exhibited a trend similar to the data from titer plates (*Figure 8E and F*). In contrast, as shown in *Figure 11*, as in the case of *E. coli* directly transformed with libraries and grown on selection plates, the trinucleotide mutation patterns obviously differed between the non-transcribed and transcribed strand of the *supF* gene. Although the trinucleotide mutation patterns were observed to be biased between the strands, this is assumingly due to the phenotypic variability among the mutation types and mutation sites in the *supF* gene. The combined data of the trinucleotide substitution patterns from both strands becomes similar to that from the titer plates. Remarkably, the substantial amount of data on mutation spectra that we obtained had led to a better appreciation of the fact that UV-induced DNA lesions had resulted in mutation clusters with a significant strand bias. As mentioned above, the rates of N$_{12}$-BC sequences with multiple mutations were increased in a UV dose-dependent fashion (*Figure 10A and B*), therefore we next focused on the correlation between the numbers, types, positions, and mutated bases for each mutation. At first, the N$_{12}$-BC sequences were classified as containing a single mutation versus two or more mutations, and N$_{12}$-BC sequences in each category were classified by the type of mutations they contained (*Figure 12A*). The total number of N$_{12}$-BC sequences with mutations is indicated in brackets. If an N$_{12}$-BC sequence contained one mutation, the proportion of mutations classified as Del-ins or SNSs from cytosine sites were increased in a dose-dependent manner. On the other hand, if an N$_{12}$-BC sequence contained more than one mutation, the mutations classified as DNSs and SNSs from guanine sites were increased. Then, the N$_{12}$-BC sequences were classified as containing 1 to 4~ mutations per sequence, and depicted in *Figure 12—figure supplement 1*, where the mutations are shown according to their nucleotide positions. It can be seen that if an N$_{12}$-BC sequence contained one mutation, especially in the case of SNSs, the mutations were detected at relatively specific sites that can be considered as sites responsible for *supF* mutation phenotype. On the other hand, if an N$_{12}$-BC sequence contained more than one mutation, the mutations were relatively sparsely distributed. In both cases, SNSs were predominantly substituted from either C or G. This result may suggest that the DNA lesions produced by UV irradiation at cytosine sites on the same strand of the DNA molecule produce multiple mutations simultaneously at the first replication cycle.

Next, the SNSs were classified as either one or multiple SNS(s) per N$_{12}$-BC sequence and shown for the non-transcribed strand of the *supF* gene with all 12 types of base substitutions (*Figure 12B*). In the case of single SNSs (*Figure 12B*, graph on top), G to A was the most frequent and C to T was the second most frequent substitution, which may be associated with the number of G bases (18 sites) and C bases (14 sites) in the strand of the *supF* gene responsible for the selection phenotype (refer to Figure 14A). On the other hand, in the case of multiple SNSs per N$_{12}$-BC sequence (*Figure 12B*, bottom graph), C to T was the most frequent and G to A was the second most frequent, which may be associated with the number of G bases (56 sites) and C bases (70 sites), GA (13 sites) or TC (18 sites) in the non-transcribed strand of the *supF* gene (refer to Figure 14A and B ). The number of N$_{12}$-BC sequences with other types of single mutations are shown in *Figure 12C*, where a clear UV dose-dependent increase can be seen in the number of 1-nt deletions, DNSs, and Del-ins, but not in

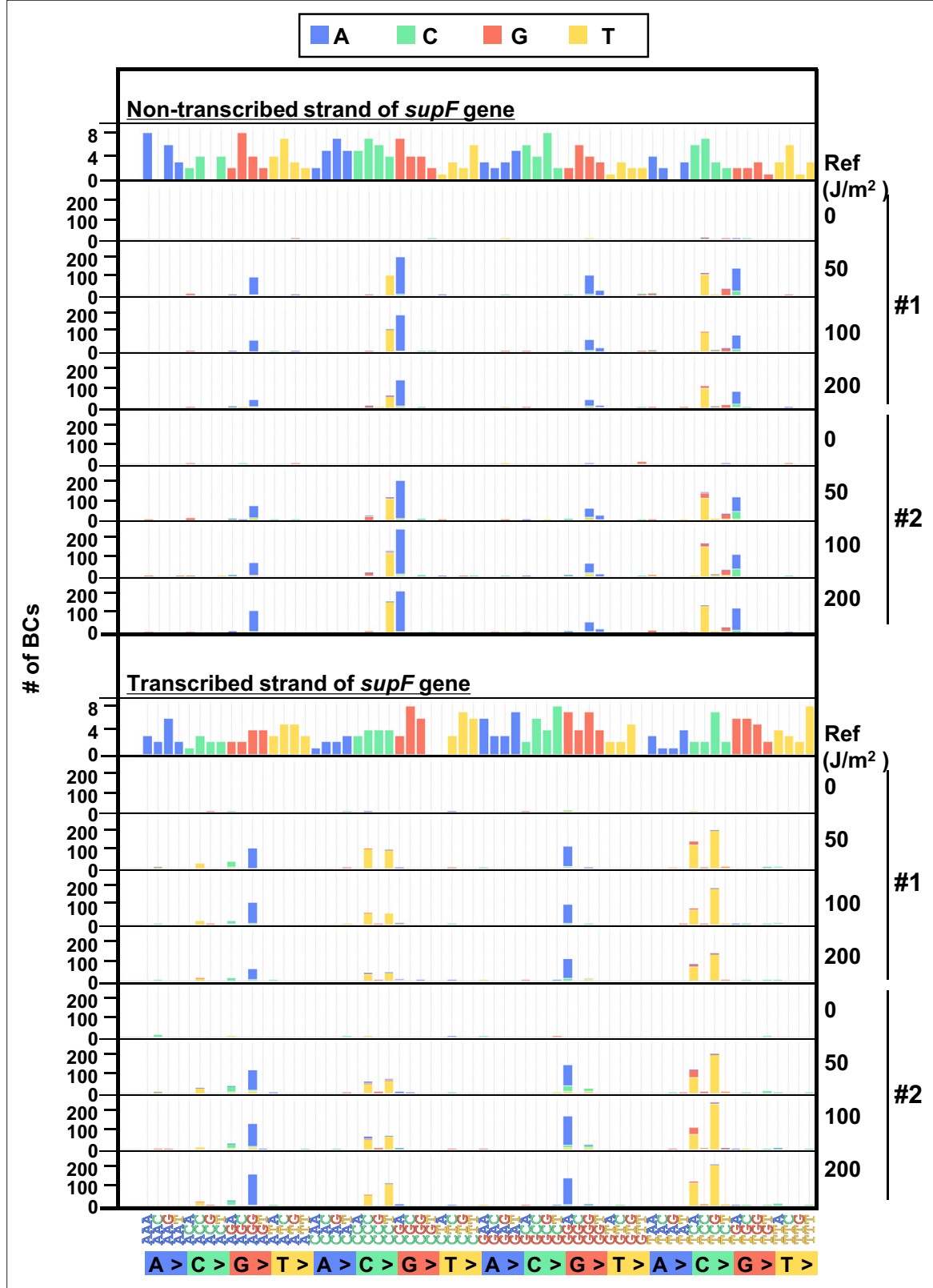

**Figure 7.** The 192 trinucleotide contexts of ultraviolet (UV) radiation induced single nucleotide substitutions (SNSs) in bacterial cells (data from selection plates). SNSs induced by UV irradiation in the *supF* gene in 192 trinucleotide contexts. The strand-specific number of BCs are shown for each SNSs, analogous to *Figure 5I*, but the data are independently plotted for libraries Lib10⁴_#1 and Lib10⁴_#2. Source data is available in *Supplementary file 4E*.

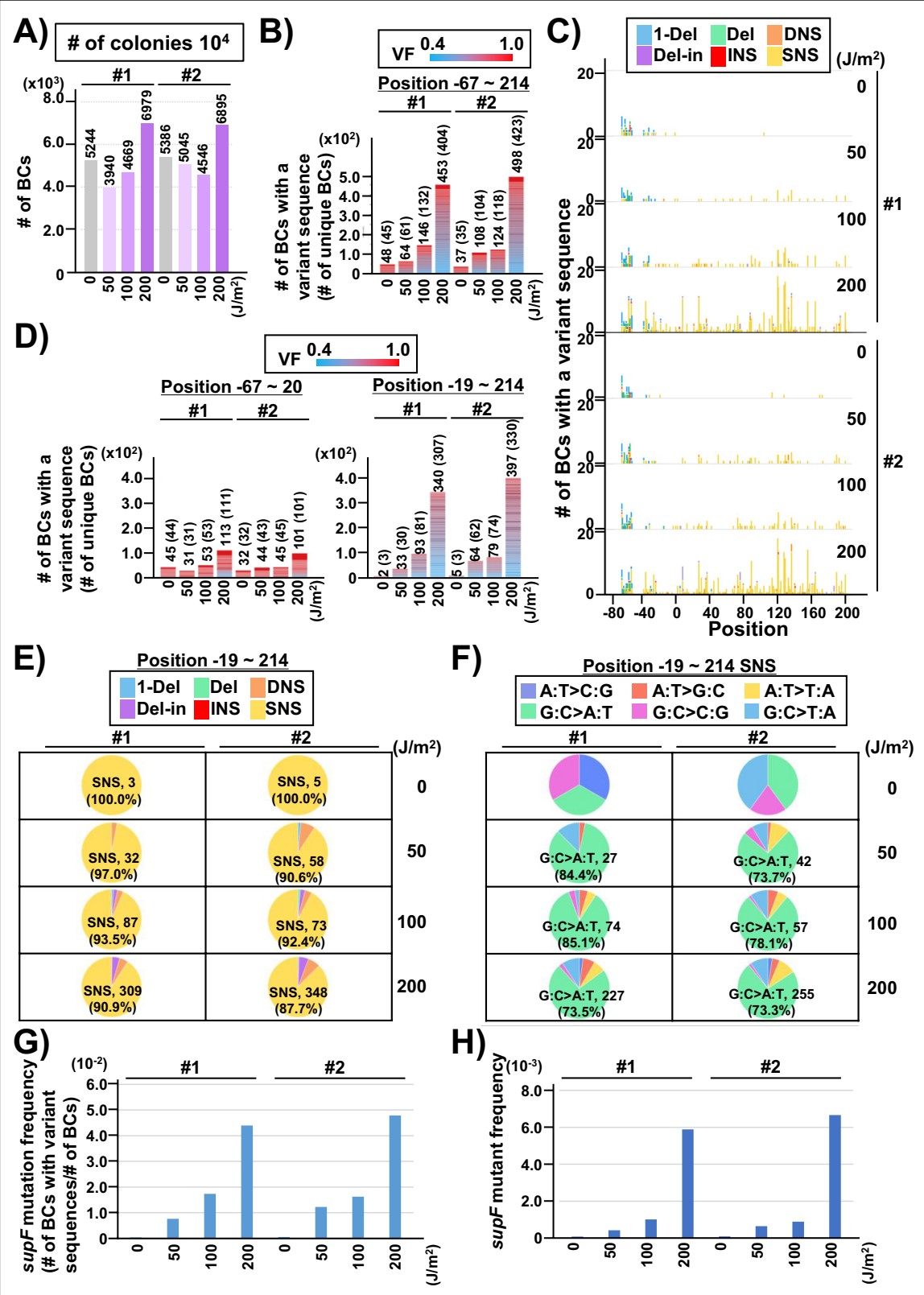

**Figure 8.** Performance of the *supF* mutagenicity next-generation sequencing (NGS) assay in mammalian cells (data from titer plates). The layout of the figure is analogous to *Figure 4*, this time for RF01 transformed with libraries extracted from transfected U2OS cells. (**A**) Numbers of classified N12-BC sequences analyzed from approximately 10^4 colonies from titer plates. Source data is available in *Supplementary file 5A*. (**B**) Number of N12-BC sequences with a variant sequence exceeding variant frequency (VF) 0.4 (nucleotide positions from –67 to 214). (**C**) Number of N12-BC sequences

*Figure 8 continued on next page*

*Figure 8 continued*

with a variant sequence exceeding VF 0.4 and their distribution by nucleotide position. (**D**) The dataset shown in panel B is now separated into two graphs according to nucleotide position relative to the *supF* gene. The graph on the left side represents positions from –67 to –20, and the graph on the right side represents positions from –19 to 214. (**E**) Pie charts of the percentage distribution of different mutation types. Source data is available in *Supplementary file 5B*. (**F**) Pie charts of the percentage distribution of different single nucleotide substitutions (SNSs). Source data is available in *Supplementary file 5C*. (**G**) The *supF* mutation frequencies determined by the NGS assay. (**H**) The *supF* mutant frequencies determined by the conventional method.

The online version of this article includes the following figure supplement(s) for figure 8:

**Figure supplement 1.** Number of $N_{12}$-BC sequences with a variant exceeding variant frequency (VF) 0.4 at different UV-C doses and their distribution by nucleotide position (in mammalian cells, data from titer plates).

---

the number of large deletions and insertions. *Figure 12D and E* represents the most frequent mutation sequences for DNS and Del-in mutations, respectively. The DNSs were gradually increased in a dose-dependent manner and seem to be saturated at a dose of 200 J/m$^2$; also, they seem to mainly result from GG:CC to AA:TT transition in at least tetra-nucleotide repeats of C or T at several distinct positions in the gene (refer to Figure 14C). On the other hand, in the case of Del-ins, the mutations increased at higher doses and were the result of GAG:CTC to AAA:TTT transitions, also at distinct positions (refer to Figure 14D ). These triplet mutations may be generated from UV photolesions at TC dipyrimidine sites by error-prone TLS polymerases such as pol ζ and polκ. In addition, in the non-irradiated samples the Del-in mutations were peculiarly different from those in the irradiated samples and were detected as TTGAT to ATCAA. This TTGAT penta-nucleotide sequence located in the potential hairpin loop of a quasi-palindromic sequence, GGCGCGTCATTTGATATGATGCGCC, may lead to template switching and give rise to complementary cluster mutations (refer to Figure 14E).

Next, we focused on the multiple mutations (more than one mutation in the same $N_{12}$-BC sequence) in terms of their combinations and mutation spectra. The multiple mutations were classified by their combinations and depicted for each UV dose in *Figure 12—figure supplement 2*. The comparison between the spectra of the single and multiple mutations for the sample irradiated with 200 J/m$^2$ was summarized in *Figure 13* (single mutation per $N_{12}$-BC – *Figure 13*, bottom part; more than one mutation per $N_{12}$-BC – *Figure 13*, top and center-left part) and *Figure 13—figure supplement 1*. It is obvious that there is a consensus for multiple mutations in a single $N_{12}$-BC sequence as follows: (1) In the non-irradiated samples the mutations were by a large majority SNSs, all of them either from C or G (*Figure 12—figure supplement 2*). (2) In the UV-irradiated samples all combinations of more than two mutations in a single $N_{12}$-BC sequence were increased in a dose-dependent manner, except for those composed only of substitutions from A and T. (3) As for the Del-in and DNS mutations, if they contained C to T substitutions, the other mutation(s) were mainly C to T SNSs, and if they contained G to A substitutions, the other mutations were mainly G to A SNSs (*Figure 13—figure supplement 1*). (4) The number of $N_{12}$-BC sequences with multiple mutations including a mutation from C was higher than those including a mutation from G, same as the trend for multiple SNS mutations shown in *Figure 12B*. (5) The distance between two SNSs in multiple mutations induced by UV irradiation was relatively shorter than the theoretically expected based on the sequence (*Figure 13—figure supplements 2 and 3*).

## Discussion

Ever since the 1980s, the *supF* forward mutagenesis assay which utilizes a *supF* gene encoding shuttle vector and an indicator *E. coli* has been widely employed in studies on mutagenicity in *E. coli* or mammalian cells (*Yoon et al., 2019*; *Camakaris and Pittard, 1973*; *Kraemer and Seidman, 1989*; *Seidman et al., 1985*; *Yoon et al., 2021b*). This conventional *supF* assay is very useful for analysis of both mutation frequencies and mutation spectra. The mutation spectra of single or tandem base substitutions for inactive *supF* genes identified by using the blue-white colony color assays were comprehensively summarized in an earlier review article, and it was noted that the mutations were detected at 86 sites within a 158 bp region covering the *supF* gene (54%) and at 74 sites within the 85 bp mature tRNA region (87%), thus demonstrating the great sensitivity of the *supF* assay system for analysis of mutation spectra (*Kraemer and Seidman, 1989*). However, obtaining reliable datasets by the conventional *supF* assay requires skill and experience, especially for studies where the mutations

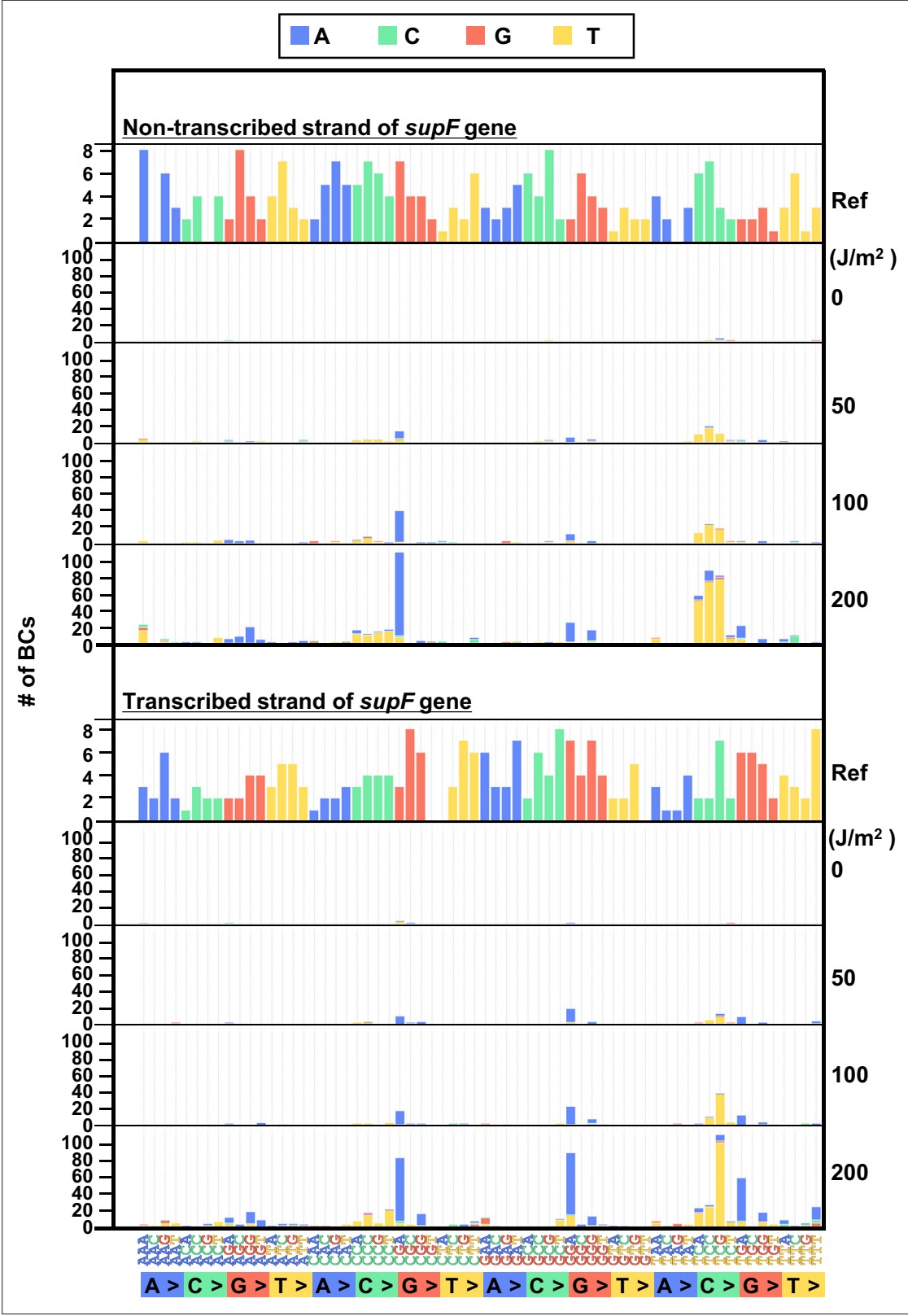

**Figure 9.** The 192 trinucleotide contexts of ultraviolet (UV) radiation-induced single nucleotide substitutions (SNSs) in mammalian cells (data from titer plates). UV-induced SNSs in 192 trinucleotide contexts for the non-transcribed (top panel) and transcribed (bottom panel) strand of the *supF* gene. Source data is available in **Supplementary file 5D**.

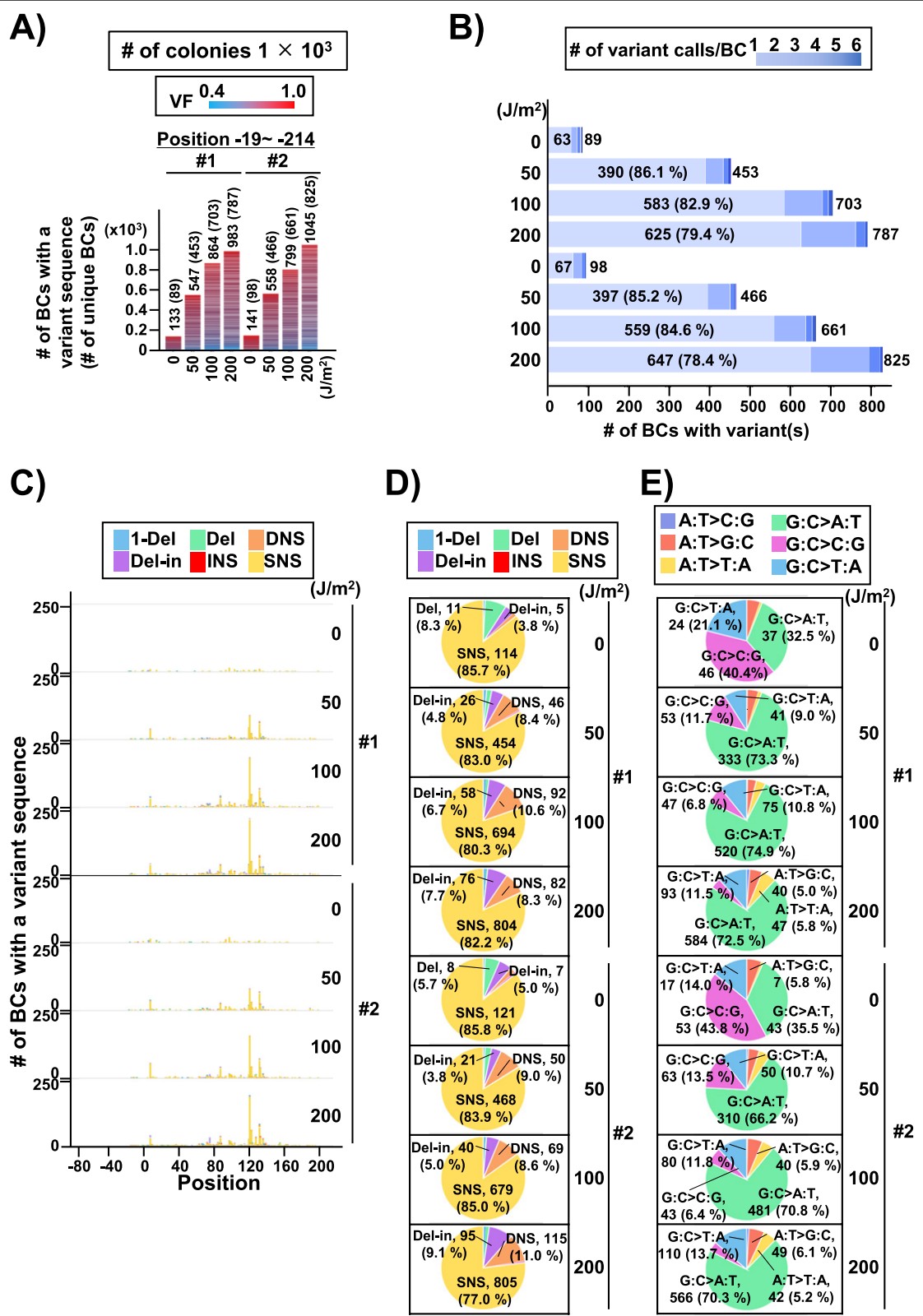

**Figure 10.** Performance of the *supF* mutagenicity next-generation sequencing (NGS) assay in mammalian cells (data from selection plates). The data represents RF01 transformed with libraries extracted from transfected U2OS cells. For identical N₁₂-BC sequences with variants the data is obtained from duplicate PCR samples. (**A**) Number of classified N₁₂-BC sequences detected from colonies in selection plates (see *Figure 1C* (iv)). Approximately 1×10³ colonies were harvested and analyzed. The number above each bar represents the number of classified N₁₂-BC sequences, and the number in

*Figure 10 continued on next page*

*Figure 10 continued*

the brackets refers to the number of unique BCs with variant sequences. Source data is available in **Supplementary file 6A**. (**B**) Number of $N_{12}$-BC sequences with single or multiple (1–6) mutations. The number of BCs with a single mutation is indicated inside each bar, with their percentage of the total $N_{12}$-BC sequences added in brackets; the total number of mutations is indicated outside the bars and corresponds to the '# of unique BCs' in panel (**A**). Source data is available in **Supplementary file 6B**. (**C**) Number of $N_{12}$-BC sequences with a variant exceeding variant frequency (VF) 0.4 according to their nucleotide position. (**D**) Pie charts of the percentage distribution of mutation types. Source data is available in **Supplementary file 6B**. (**E**) Pie charts of the percentage distribution of different single nucleotide substitutions (SNSs). Source data is available in **Supplementary file 6D**.

The online version of this article includes the following figure supplement(s) for figure 10:

**Figure supplement 1.** Number of $N_{12}$-BC sequences with a variant exceeding variant frequency (VF) 0.4 at different UV-C doses and their distribution by nucleotide position (in mammalian cells, data from selection plates).

of interest are induced with low frequency. The method has been advanced by the construction of indicator bacterial strains with different *supF* reporter genes which allow selection based on survival of bacteria containing mutant *supF* genes. However, the fact that the *supF* phenotypic selection process relies on the structure and function of transfer RNAs that may be differently affected by different mutations means that the improvement of the efficiency of the selection process may cause loss of coverage of the mutation spectra, as it is under our experimental conditions, where the coverage is about 30% (**Kraemer and Seidman, 1989**; **Fukushima et al., 2020**). Since in recent years NGS has become a routine research tool due to its lowering cost, numerous sophisticated applications of NGS technologies are being developed for research in genomics, epigenomics, and transcriptomics (**McCombie et al., 2019**; **Wang et al., 2016**). In this study, we attempted to combine the *supF* shuttle vector forward mutagenesis assay with sequencing analysis by a short-read NGS platform and consequently developed a novel *supF* NGS assay with potential application in mutagenesis and DNA repair studies. The combination of the DNA-barcoded *supF* shuttle vector library and its amplification via in vivo replication in the indicator non-SOS *E. coli* provides us with proper templates for the PCR-facilitated preparation of multiplexed NGS samples. Thus, during the PCR process the amplification and index tagging of the samples are achieved simultaneously. Designed in such a way, the assay allows us to obtain almost all mutation sequences from a small amount of data from NGS reads, especially in samples from colonies grown on selection plates, where the mutation sequences, although prone to bias, are highly concentrated. This results in the possibility to simultaneously analyze hundreds of samples from different experiments in a single NGS run. Here, the cost of mutation spectrum analysis from selection plates was less than $1.00 per sample, even though both the NGS library preparation and the NGS run were outsourced. Further cost reductions can be expected by obtaining more sequencing data in a single run, and also from the continuous improvement of NGS technologies. From this point of view, we believe that we can secure a sufficient number of experiments to improve the accuracy of the analysis and to confirm the reproducibility of the experiments. Furthermore, the data from colonies grown on titer plates provides us, at least in principle, and with the exception of large deletions and insertions, with non-biased mutation spectra for the whole analyzed region. Also, mutation frequencies calculated from titer plate data were 15–20 times higher than those obtained by the conventional *supF* assay, demonstrating that our novel assay allows us to analyze more thoroughly the mutation spectrum by using a smaller amount of data and to detect mutations with lower frequencies. The analysis of data from both selection and titer plates is consistent with the desired reliability and low cost of the mutagenesis assay. Last but not least, by using the *supF* shuttle vector plasmid, mutation spectrum data can be obtained with different experimental designs, such as pre-treatment with metabolic enzymes or using cells with different genetic backgrounds, siRNA transfection, CRISPR-Cas systems, etc. We are currently conducting experiments with an *supF* library containing site-specific single DNA damages such as 8-oxo-7,8-dihydroguanine (8-oxoG, 8-hydroxyguanine) among others (**Suzuki et al., 2018**; **Suzuki et al., 2021**; **Suzuki et al., 2022**), which will allow us to analyze in detail the repair mechanisms of sequence-specific DNA damage and provide further insights into mutagenesis and DNA repair processes.

The current study posits that when performing the assay, we have to be aware of the following points: (1) The required barcode complexity defined by the number of barcode sequences in the *supF* shuttle vector library depends on the experimental purpose and the mutation detection frequencies. A barcode complexity of $1.0 \times 10^4$ for each library is used as the standard in this paper. Even though more complexity may be able to provide better results, especially at lower mutation frequencies, one

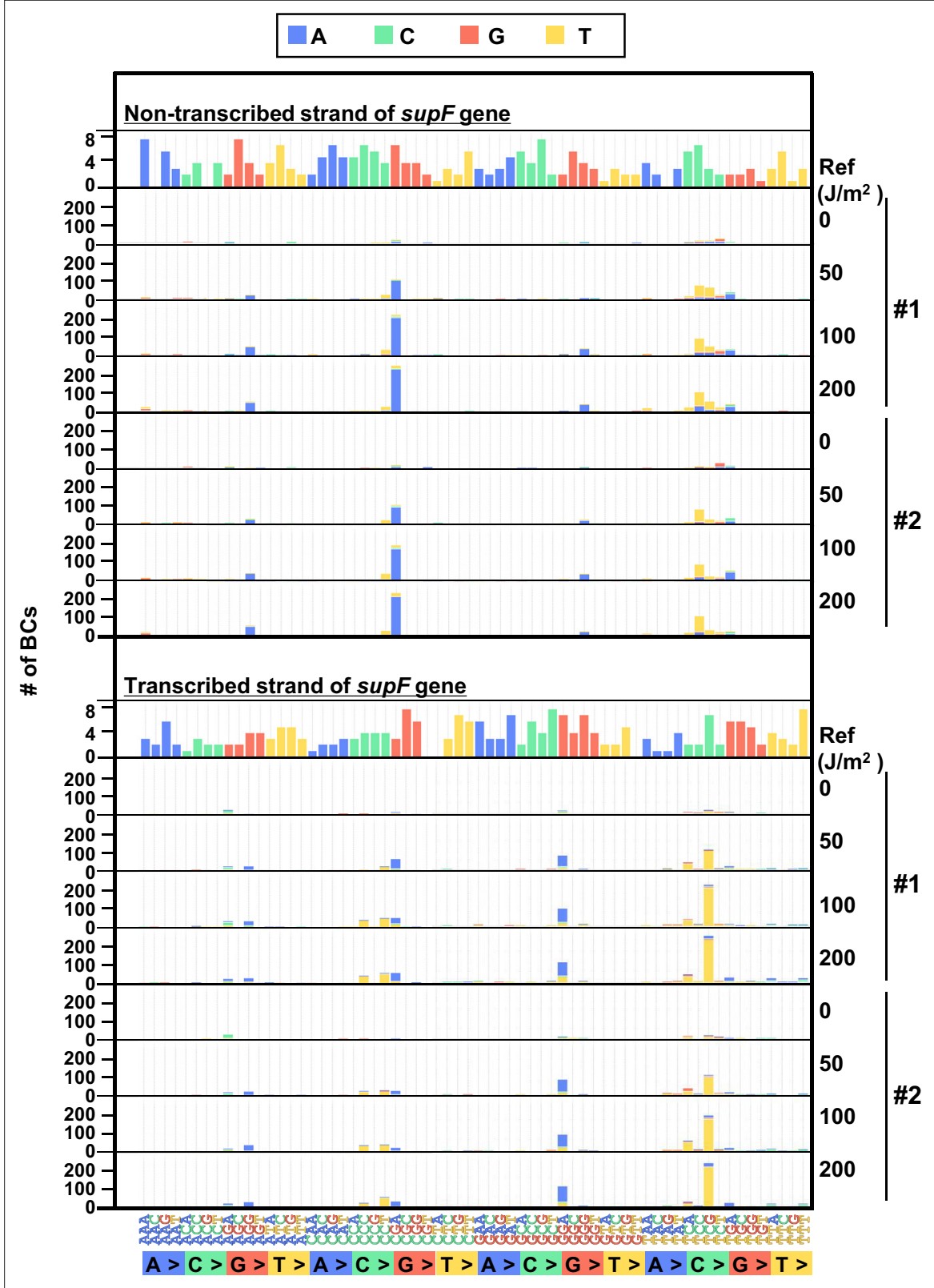

**Figure 11.** The 192 trinucleotide contexts of ultraviolet (UV) radiation-induced single nucleotide substitutions (SNSs) in mammalian cells (data from selection plates). SNSs induced by UV irradiation in the *supF* gene shown in 192 trinucleotide contexts. The data for the non-transcribed (top panel) and transcribed (bottom panel) strand of the *supF* gene is shown. Source data is available in *Supplementary file 6E*.

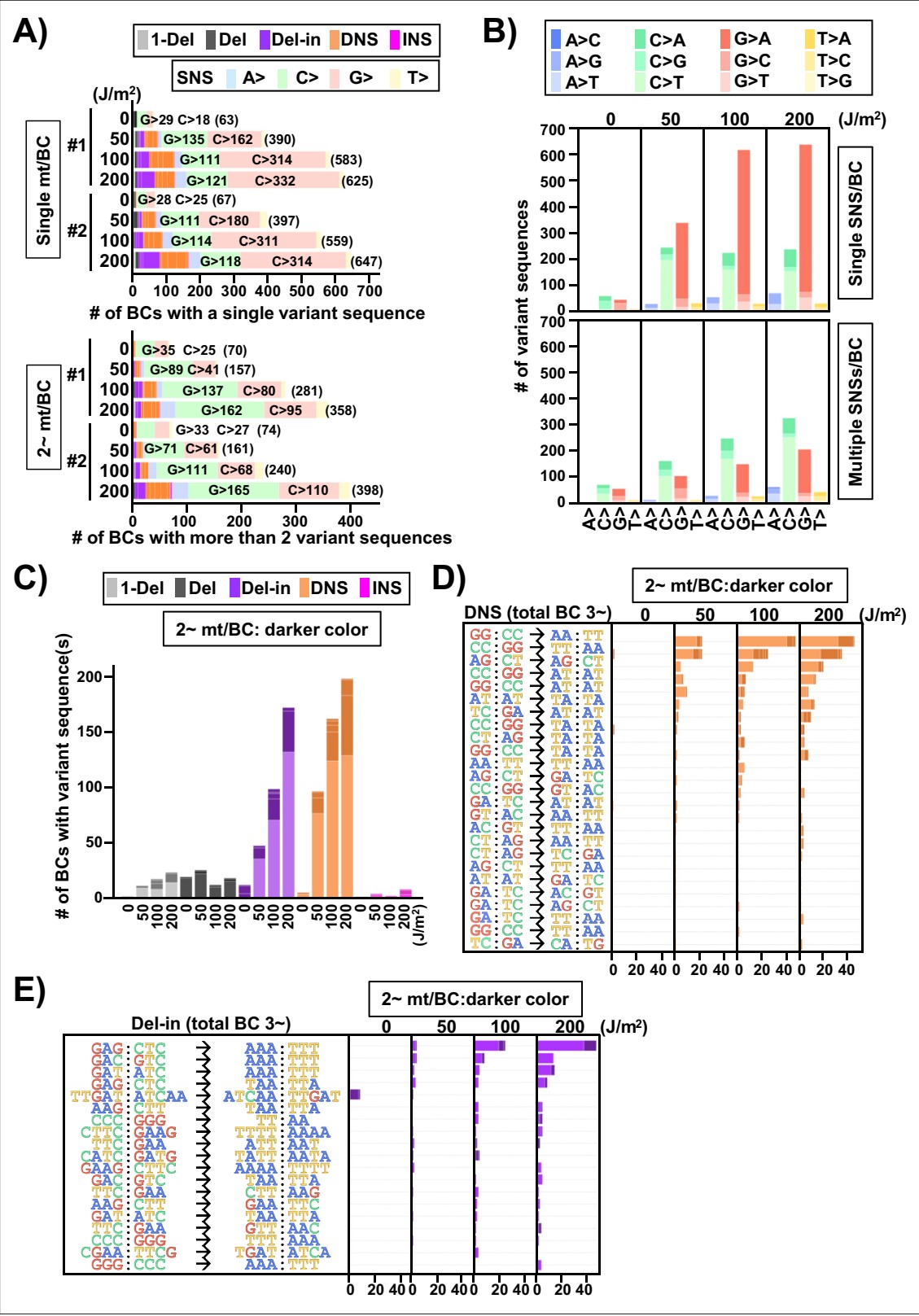

**Figure 12.** Ultraviolet (UV)-induced mutations in the *supF* gene in mammalian cells (data from selection plates). The mutation sequence on the non-transcribed strand of the *supF* gene is analyzed. (**A**) Proportions of mutation types in BCs with a single variant sequence (single mt/BC) or more than one variant sequence (2~ mt/BC). Source data is available in ***Supplementary file 7A***. (**B**) Number of single nucleotide substitutions (SNSs) induced by UV irradiation – one SNS per BC (graph on top), and multiple SNSs (2~) per BC (bottom graph). The nucleotide sequences of SNSs are indicated

*Figure 12 continued on next page*

*Figure 12 continued*

by colors as shown in the legend. Source data is available in *Supplementary file 7B*. (**C**) Number of mutations induced by UV irradiation classified by mutation type, except for SNSs. The mutation types are shown in different colors as indicated in the legend. For each color, the lighter shade indicates single mutation per BC, and the darker shade indicates multiple mutations per BC. Source data is available in *Supplementary file 7C*. (**D**) Number of dinucleotide substitutions (DNSs) and their mutation sequences. The mutations have been listed if more than two identical mutation sequences among all samples have been detected. Source data is available in *Supplementary file 7D*. (**E**) Number of deletions with insertions (Del-in) mutations and their sequences. The mutations have been listed if more than two identical mutation sequences among all samples have been detected. Source data is available in *Supplementary file 7E*.

The online version of this article includes the following figure supplement(s) for figure 12:

**Figure supplement 1.** The position and mutation type of each identified mutation (data from selection plates, 0–200 J/m$^2$).

**Figure supplement 2.** The combinations of ultraviolet (UV)-induced mutation types in the supF gene in mammalian cells (data from selection plates, 0–200 J/m$^2$).

must keep in mind that the number of colonies to be harvested for the NGS sample preparation, as well as the platform or setting for data analysis, need to be adjusted depending on the number of barcodes. (2) The number of NGS reads required for each multiplexed sample and for each $N_{12}$-BC sequence need to be calculated before the preparation of the samples. Therefore, the number of $N_{12}$-BC sequences and the number of harvested colonies must be optimized first. In addition, the allele frequency given by VarDict, referred to as VF in this study, for each classified $N_{12}$-BC sequence is the most important determinant of the quality of the variant sequence data and the basis for analyzing overlapping $N_{12}$-BC sequences. The cost, time, and data quality must be all well considered and balanced when an experiment is designed. (3) NGS data contain sequencing errors, the rate of which is known to be between 10$^{-3}$ and 10$^{-4}$, with the majority of the errors being A:T to G:C and G:C to A:T transitions that had occurred during the PCR in the process of NGS library preparation (*Brodin et al., 2013*; *Costello et al., 2013*). In the assay presented here no sonication step is required, thus avoiding DNA oxidation. As already mentioned, the clonal amplification of the *supF* shuttle vector via in vivo replication in the non-SOS *E. coli* provides a proper PCR template for preparation of NGS samples. In addition, the sequence alignment to the reference sequence for all identical $N_{12}$-BC sequences and the analysis of data obtained from duplicate PCR reactions can almost completely eliminate the sequencing errors resulting from the process of sample preparation. On the other hand, in the case of the amplicon sequencing employed here, the base-calling errors occurred depending on the 'per base sequence quality' of the specific sequence. In this study, in addition to the *supF* gene with its total length of 335 bp being sequenced, 200 bp paired-end sequencing was performed with about 30 bp overlapping sequences, and the resulting sequence was merged together. The 'per base sequence quality' in the region between read positions 160 and 200 bp is relatively low compared to other positions. Data shown in this study, as well as other experiments conducted in our laboratory, reveals that two positions in particular – 55 and 69 in the *supF* gene manifest a very high frequency of variant calls – 20–50% of samples, but with low VF (less than 0.3). These two positions are located at 165–167 (depending on the $N_0$, $N_1$, $N_2$ primers) and 181–183 bp from the end of the forward reads, and at 179–181 and 167–169 bp from the end of the reverse reads. The addition of different numbers of random nucleotides – 0-, 1-, or 2-nt at the end of both primers used for sample preparation can contribute to better 'per base sequence quality' in clonal amplicon sequencing. However, positions 55 and 69 are located immediately next to CC:GG sequences (position 55 is located in TTA̲CC:GGT̲AA and position 69 is located in GTT̲CC:GGA̲AC), and the result of variant calling was almost exclusively SNS from 55A:T or 69T:A to C:G. Thus, these variant calls appear to be position-specific and sequence-dependent, and for this assay can be regarded as position-specific base-calling errors. Since they had a VF of less than 0.3 and a threshold VF of 0.4 was applied in this study, these base-calling errors do not need to be given consideration. Nevertheless, it must be emphasized that fine-tuning must be done for each individual experiment. (4) Large alterations in the *supF* gene, such as a large deletion or insertion, cannot be analyzed by this assay because of impossibility for PCR amplification or read alignments. This may render our observations on mutation frequency misleading. The frequency of such large alterations is better assessed by colony PCR, especially if the number of colonies grown on the titer plates was different from expected. (5) For the *supF* assay, spontaneous cluster mutations at TC:GA sites were often observed, and it was well illustrated in an earlier study that a nick in the shuttle vector was a trigger for these asymmetric cluster mutations (*Seidman et al.,*

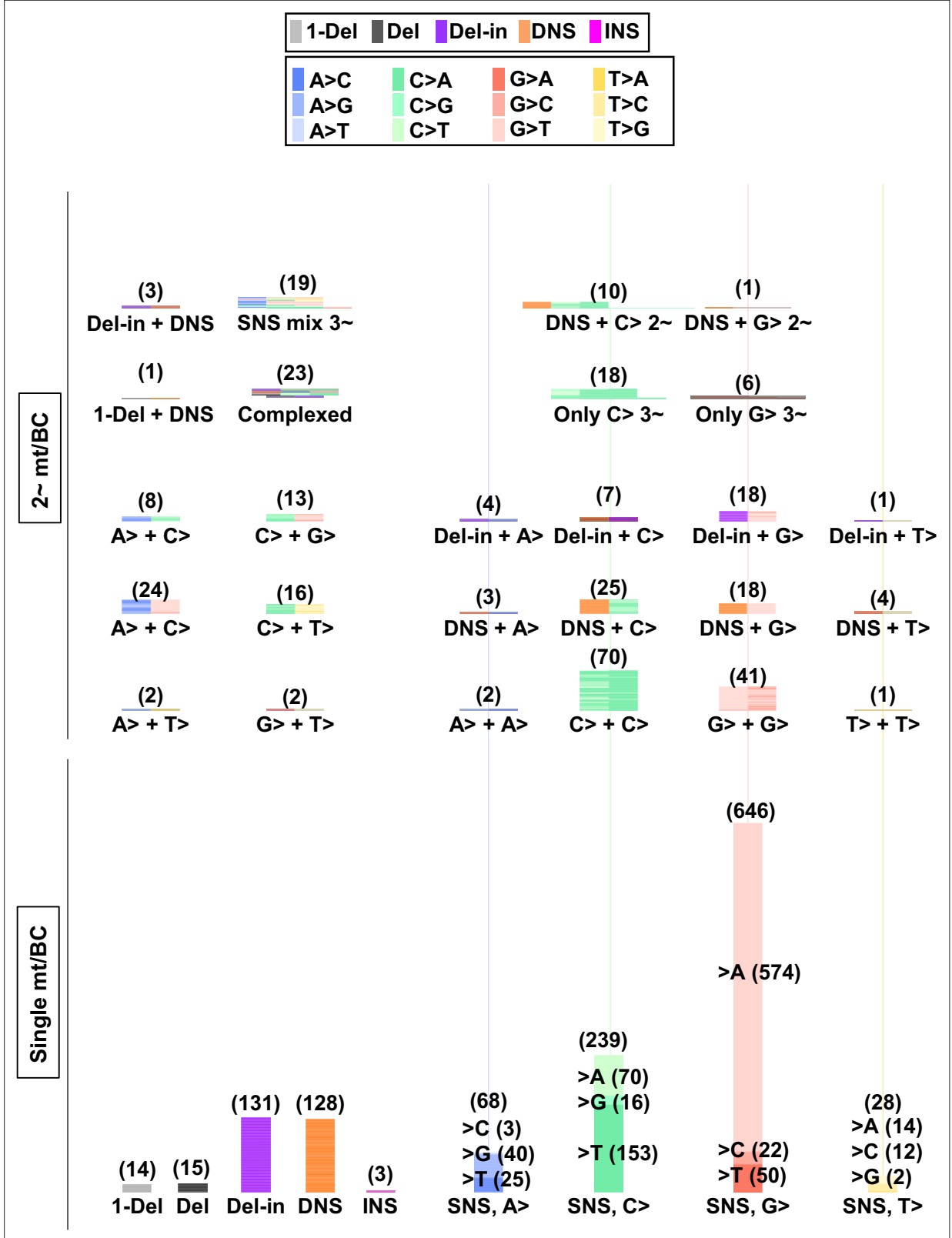

**Figure 13.** Numbers and combinations of detected mutations per N$_{12}$-BC in mammalian cells (data from selection plates, 200 J/m$^2$). The numbers of detected mutations for each single mutation per N$_{12}$-BC (lower part) and for combinations of more than one mutations per N$_{12}$-BC (upper and middle part) are shown by stacked thin horizontal bars. The type of mutation is indicated by a different color (legend on top) and by its abbreviation on the horizontal axis. The number in brackets and the height of the stacked horizontal bars represent the number of BCs with the mutation(s).

*Figure 13 continued on next page*

*Figure 13 continued*

The online version of this article includes the following figure supplement(s) for figure 13:

**Figure supplement 1.** The combinations of ultraviolet (UV)-induced mutation types and mutation spectrum in the supF gene in mammalian cells (data from selection plates, 200 J/m²).

**Figure supplement 2.** The distance between multiple single nucleotide substitutions (SNSs) per BC (data from selection plates, 50, 100, and 200 J/m²).

**Figure supplement 3.** The distribution of the distance between two single nucleotide substitutions (SNSs) per BC (data from selection plates, 50, 100, and 200 J/m²).

---

*1987*). Therefore, we need to be aware of the quality of each library and how it affects the outcome of each analysis, especially for detection of very low levels of mutations. Depending on the purpose of the experiments, in the preparation of covalently closed circular vector libraries, it is essential to eliminate the background level of mutations. In fact, the in vitro construction of the library of double-stranded shuttle vectors from single-stranded circular DNA requires the process of treatment with T5 exonuclease, which drastically decreases background mutations.

In this study, to test the performance of our newly developed *supF* NGS assay, we conducted experiments using $N_{12}$-BC libraries irradiated with UV-C at different doses. Although these pilot experiments were designed in a relatively simple way, they clearly exemplify the mutation spectrum associated with UV radiation exposure, and further experiments may provide profound insights into UV mutagenesis. Here, by using this novel assay we were able to confirm in both *E. coli* and human cells that the SNSs induced by UV were mainly C:G to T:A transitions, which is a known UV signature, and that the UV-induced DNA lesions were basically dipyrimidines – CPDs at T-T sites, and (6-4)PPs at T-T and T-C sites (*Varghese and Wang, 1967*; *Mao et al., 2016*; *Lindberg et al., 2019*; *Kamiya et al., 1993*; *Kamiya et al., 1998*). The SNSs are listed in *Figure 10B* (presented within pentanucleotide sequences of the *supF* gene) starting from the most frequent. In samples obtained from titer plates, the nucleotide sequence in eight of the top ten mutational hotspots was TT**C**/**G**AA, and the sequence of the top four mutational sites was TT**C**GA/TC**G**AA. These top four sites represent two palindromic TTCGAA/TTCGAA hexanucleotide sequences located at positions 115–120 and 123–128 in the *supF* gene. All top 10 mutations were C to T transitions and occurred at T**C** sites, and are likely the result of TLS opposite the 3'C of CPDs and (6-4)PPs via specialized DNA polymerases, such as Pol $\eta$, -$\iota$, -$\zeta$, -$\kappa$, -θ, -$\lambda$, and Rev1 (*Achar and Foiani, 2019*; *Yoon et al., 2021a*). Notably, the site selection bias observed for the top 10 mutation hotspots seems to differ between the titer and the selection plates. In selection plates, three of the top ten hotspot mutations were C to T transitions at the 3'C of CC. These hotspot SNSs, particularly positions 130 and 121, appear to be next to or between hotspot SNSs in the titer plates. However, such SNS mutations, that were only observed in selection plates, were identified in both titer and selection plates as other types of mutations, specifically as hotspot DNS mutations, CC to TT, as shown in *Figure 10C* (located at 130 and 131 TTCGAAT**CC**TT, at 121 TTCGAA**GG**TTCGAA, at 5 TTT**GA**TA/T**AT**CAAA, or at 84 and 85 AA**GGG**AG/CTC**CC**TT). In addition, as shown in *Figure 9B and C*, the multiple mutations, either C or G mutations at the same base in identical $N_{12}$-BC sequences, were identified in a dose-dependent manner more frequently from selection plates where TLS polymerases opposite UV lesions on the same DNA strand can cause clustered mutations. These multiple mutations include at least one phenotypic mutation in the *supF* gene but include also non-phenotypic mutations, which contributes to the specifics of the mutation spectrum in the selection plates. The two-base substitutions within trinucleotides, referred to as Del-in by VarDict, were mainly detected at higher UV doses as G**AG**:C**TC** to A**AA**:**TT**T at positions 73–75. It is unclear whether these mutations arose through unexpected UV-induced lesions at G**C**TCG sites, or the deamination of either the 5'C or the 3'C has occurred at these specific sites by different mechanisms. Alternatively, it can be hypothesized that the UV-induced DNA lesions arose on a single-stranded DNA (ssDNA) followed by a blockage of replication, or during DNA repair processes. It is known that ssDNA can be a substrate for a subclass of activation-induced deaminase/apolipoprotein B mRNA-editing catalytic polypeptide-like subfamily cytosine deaminases (APOBECs), which cause cluster mutations at TC:GA sites in the same strand of DNA by converting cytosine to uracil (*Pecori et al., 2022*). It is realistic to assume that the mutations at TC:GA sites had occurred via several different mechanisms. In fact, spontaneous mutations in non-UV-irradiated samples transfected into U2OS cells can also be detected at the order of $10^{-4}$ per gene. In both UV-irradiated and non-irradiated samples,

SNS mutations had basically occurred from C:G at T<u>C</u>:<u>G</u>A sites. Also, the major proportion of multiple mutations were C:G to T:A transitions. However, in non-UV-irradiated samples, the major proportion of single mutations were C:G to G:C transversions, which differed from the irradiated condition, where the major proportion was G:C to A:T transition (*Figure 9C*). It is possible that the oxidatively damaged base 8-oxoG was endogenously generated in the U2OS cells. As we have previously reported, 8-oxoG becomes a trigger for the DNA repair process and generates ssDNA as a substrate for APOBECs (*Suzuki et al., 2022*). Here, the TT<u>C</u>/<u>G</u>AA site has emerged from data from titer plates as a mutational hotspot. According to previous reports, the TTC motif, and more specifically the cytosine in the motif, is a site of preference for APOBEC3A, APOBEC3C, APOBEC3F, and APOBEC3G, even though further investigation is needed (*Lindberg et al., 2019*; *Love et al., 2012*; *Adolph et al., 2018*).

What might be more, a particular mutation was detected by VarDict in non-UV-irradiated samples as a Del-in mutation – a change from TT<u>GAT</u>:<u>ATC</u>AA to <u>ATC</u>AA:TT<u>GAT</u> at positions 3–7 within the potential hairpin loop of a quasi-palindromic sequence (*Figure 14E*), which may be the consequence of template switching events. The mutation was detected with a frequency at the order of $10^{-6}$ per gene (*Figure 12E*), and was hardly detected in UV-irradiated samples, where it remains hidden by the UV-introduced mutations. Also, the same mutation was frequently detected as single SNS mutations from T:A to A:T at position 3, from G:C to C:G at position 5, and from T:A to A:T at position 7, or as combinations with other SNS mutations either or both from G:C to A:T at position –3 or/and from T:A to C:G at position 12 (*Figure 12—figure supplement 1*), which may be dependent on the regions used for template switching. Although the mono- and poly-ubiquitination of PCNA, facilitated by ubiquitin ligases Rad6-Rad18, HLTF, or SHPRH, is involved in the mechanism of switching between TLS and template switching in mammalian cells, the details are still ambiguous (*Chiu et al., 2006*; *Lin et al., 2011*; *Masuda et al., 2012*; *Wong et al., 2021*). In addition, our recent studies on action-at-a-distance mutations, according to which a site-specific targeted single 8-oxoG or other types of DNA damage cause cluster mutations at untargeted positions, demonstrated a drastic strand bias for mutations at cytosines, specifically at 5'-TC-3' sites (*Suzuki et al., 2018*; *Suzuki et al., 2021*; *Suzuki et al., 2022*). Data on mutational strand asymmetries from cancer genome sequencing studies have emerged, indicating that APOBEC-associated mutagenesis during replication strongly favors the lagging-strand template (*Seplyarskiy et al., 2016*; *Haradhvala et al., 2016*). And most recently, it has been proposed by a study using UPD-seq technique that APOBEC3A targets cytosines in small hairpin loops, within replication forks, in tRNA genes and at transcription start sites (*Sakhtemani et al., 2022*). In this study, strand biases in base substations from either C or G were clearly observed, consistent with what could be expected from the mechanisms by which UV photolesions are processed, such as cytosine deamination and the involvement of TLS. Interestingly, the strand bias switched between single base substitutions and cluster mutations depending on the doses of UV and the number of mutations per $N_{12}$-BC. The precise reason and mechanisms behind this remain to be elucidated, but the observation brings new insights into the mechanisms of mutagenesis. In addition, the positions of SNSs in the multiple mutations were closer to each other compared to the theoretically expected positions (*Figure 13—figure supplements 2 and 3*), which may reflect switching events involving TLS polymerases. It should be noted that the presented data for the distance between two SNSs in the multiple mutations was analyzed from the data from selection plates in order to secure a sufficient number of mutations, and therefore, there may be a bias due to hotspots associated with the selection process. However, the results from the limited number of mutations from the titer plates are similar to these from the selection plates. It can be proposed that this assay may also be applied for analysis of TLS polymerases in mammalian cells.

Mutational signatures identified in cancer cells are emerging as valuable markers for cancer diagnosis and therapeutics. Innumerable physical, chemical, and biological mutagens, including anticancer drugs, induce characteristic mutations in genomic DNA via specific mutagenic processes. The mutation spectra obtained here by using the presented advanced method were in good agreement with accumulated data from previous papers where the conventional method had been used, with the advantage that our method provided less-biased mutation spectra data. As described above, the datasets presented here highlighted novel mutational signatures and also cluster mutations with a strand bias, which could be associated with the processes of replication, transcription, or repair of DNA damage, including a single-strand break (a nick). In this study, eight series of *supF* shuttle vector plasmids were constructed, as presented in *Figure 1—figure supplement 1*; however, the analysis

**A)**

```
                                                    *     *    *        *
                                         CACTTTACA GCGGCGCGTC  -1

       -19

        *   *           *
   1  ATTTGATATG ATGCGCCCCG CTTCCCGATA AGGGAGCAGG CCAGTAAAAG 50
          ❹                        ⑦

                     *****  ** ** *      *  **   ** *   *** **
  51  CATTACCTGT GGTGGGGTTC CCGAGCGGCC AAAGGGAGCA GACTCTAAAT 100
                                               ⑧
                                               ❼            ❾

       *  * **        ** *  *******  *  *         *
 101  CTGCCGTCAT CGACTTCGAA GGTTCGAATC CTTCCCCCAC CACCATCACT 150
               ①②       ④③      ④
               ❻❶    ❸    ❿    ❷ ❺ ❽

                   *
 151  TTCAAAAGTC CGAAAAGATC CGGCGGCGCA GCACCTCATC CTCAGCGATA 200
       ⑥            ⑨                                      ❿

 201  TGGCAGCTTA TAAT                                       214
```

```
A:T(58,*7) C:G(70,*14) G:C(56,*18) T:A(49,*8)

TC:GA(18,*6) GA:TC(13,*6) GG:CC(15,*6) CC:GG(20,*6) TT:AA(13,*4) AA:TT(17,*6)

TCA:TGA(6, 0) TCC:GGA(7,*2) TCG:CGA(3,*2) TCT:AGA(2,*2)
TGA:TCA(2,*1) CGA:TCG(7,*3) AGA:TCT(2,*1) GGA:TCC(2,*1)

ATC:GAT(6,*2) CTC:GAG(3,*1) GTC:GAC(3, 0) TTC:GAA(6,*3)
GAT:ATC(5,*1) GAG:CTC(3,*1) GAC:GTC(2,*1) GAA:TTC(3,*3)
```

**B)**

```
① 117 TTCGA/TCGAA (55)
② 118 TCGAA/TTCGA (45)
③ 126 TCGAA/TTCGA (44)
④ 125 TTCGA/TCGAA (41)
④ 134 TTCCC/GGGAA (41)
⑥ 153 TTCAA/TTGAA (34)
⑦  24 TTCCC/GGGAA (31)
⑧  86 GGGAG/CTCCC (28)
⑨ 162 CCGAA/TTCGG (25)
⑩ 197 GCGAT/ATCGC (22)

❶ 118 TCGAA/TTCGA (971) (②)
❷ 130 ATCCT/AGGAT (414) (⑫)
❸ 121 AAGGT/ACCTT (261) (㉓)
❹   5 TTGAT/ATCAA (247) (⑭)
❺ 131 TCCTT/AAGGA (186) (㉖)
❻ 117 TTCGA/TCGAA (182) (①)
❼  85 AGGGA/TCCCT (170) (㉔)
❽ 134 TTCCC/GGGAA (102) (⑤)
❾  95 CTCTA/TAGAG  (98) (−)
❿ 126 TCGAA/TTCGA  (94) (③)
```

**C)**

```
① 134 TTCCCC/GGGGAA →TT/AA(2/4)
② 31 TAAGGG/CCCTTA →GA/TC(2/3)
② 85 AGGGAG/CTCCCT →AA/TT(3/3)
② 130 ATCCTT/AAGGAT →TT/AA(3/3)
② 137 CCCCAC/GTGGGG →AT/AT(2/3)

❶ 130 ATCCTT/AAGGAT →TT/AA(37/50) (②)
❷ 84 AAGGGA/TCCCTT →AA/TT(22/27) (−)
❷ 85 AGGGAG/CTCCCT →AA/TT(20/27) (②)
❹ 121 GAGGTT/AACCTC →AA/TT(18/22) (−)
❺ 70 TTCCCG/CGGGAA →TT/AA(14/21) (−)
```

**D)**

```
①   5 TTGATAT/ATATCAA →AAA/TTT(6)
② 73 CCGAGCG/CGCTCGG →AAA/TTT(5)
③ 184 CACCTCAT/ATGAGGTG →ACTT/AAGT(3)

❶ 73 CCGAGCG/CGCTCGG →AAA/TTT(73/84)
❷ 112 TCGACTT/AAGTCGA →AAA/TTT(25/30)
❸   5 TTGATAT/ATATCAA →AAA/TTT(20/27)
```

**E)**

```
   -8      -3      3 5 7    12    17
  GGCGCGTCATTTGATATGATGCGCGC        GGCGCATCATATCAAATGACGCGCC
  ***** *** * * **** *****    →     ***********************
  CCGCGCAGTAATAACTATACTCGCGG        CCGCGTAGTATAGTTTACTGCGCGG
```

Figure 14. Summary of results for ultraviolet (UV)-induced mutagenesis of the *supF* gene in mammalian cells. (**A**) The nucleotide sequence of the non-transcribed strand (sense strand) is analyzed (233 bp, positions −19 to 214). The nucleotides with an asterisk are associated with the *supF* mutant phenotype resulting from one single nucleotide substitution (SNS) according to our previous experimental data. The 5'-TC-3' sites are shown in green color, and the 5'-GA-3' sites are shown in red color. The nucleotide positions with a number in a circle correspond to the sequences listed in panel (**B**). A

*Figure 14 continued on next page*

*Figure 14 continued*

summary of the numbers of indicated 5' to 3' sequences from the non-transcribed strand and their complementary sequences from the transcribed strand are shown in the text box. The first number in the brackets represents the total number of the indicated nucleotide sequences, and the number with an asterisk represents the number of nucleotides associated with the *supF* mutant phenotype resulting from SNSs at the site of the underlined nucleotide. (**B**) The UV-induced SNSs at individual positions are sorted by the frequency of their detection (number of BCs) from most to least. The rank of the frequency is shown by the numbers in either a white circle referring to data from titer plates (corresponding to *Figures 8 and 9*) or a black circle referring to data from selection plates (corresponding to *Figures 10 and 11*). The SNS is denoted as an underlined nucleotide in a pentanucleotide sequence (including two nucleotides upstream and downstream of the mutation, and the sequence of the non-transcribed strand/transcribed strand). The number next to the rank is the position of each mutation, corresponding to the nucleotide sequence in panel (**A**). The number in brackets represents the number of BCs with the mutation. (**C**) Data for double nucleotide substitutions, analogous to panel (**B**). The majority of converted two-nucleotide sequences for each underlined two nucleotides in a hexanucleotide sequence is shown by arrows. The number of the tandem two-nucleotide substitutions is shown before the slash in the brackets, and the total number of the two-nucleotide substitutions – after the slash. (**D**) Data for deletion/insertion mutations, analogous to panels (**B**) and (**C**). (**E**) The potential hairpin loop of the quasi-palindromic sequence in the reference sequence (on the left side), and the mutated sequence detected from the selection plate of the non-irradiated sample (on the right side). The numbers represent the nucleotide position in the non-transcribed strand sequence of the *supF* gene. The potential Watson-Crick hydrogen bonded base pair is shown by an asterisk. The nucleotide sequences detected as SNS mutations at positions –3 and 12 are shown in orange color, and for deletion/insertion at positions 3–7 – in purple color.

was carried out using $N_{12}$-BC libraries prepared from either pNGS2-K1 (*Figures 1–4*) or pNGS2-K3 (*Figures 5–10*). The pNGS2-K1/-A1/-K4/-A4 and pNGS2-K2/-A2/-K3/-A3 vector series contain an M13 intergenic region with opposite orientations relative to the *supF* gene, which allows us to incorporate specific types of DNA damage at specific sites in the opposite strand of the vector library. Also, the pNGS2-K1/-A1/-K3/-A3 and pNGS2-K2/-A2/-K4/-A4 vector series contain the SV40 replication origin, which enables bidirectional replication and transcription, at opposite sides of the *supF* gene. Although this is still preliminary data, it is notable that the spontaneously induced mutations for the different vectors in U2OS cells were not significantly different. Therefore, the here presented mutagenesis assay with NGS, by using these series of libraries, can be applied in many different types of experiments to address both quantitative and qualitative features of mutagenesis. It is possible to design series of libraries containing DNA lesions or sequences suitable for the investigation of specific molecular mechanisms, such as TLS, template switching, and asymmetric cluster mutations. We are currently trying to develop a different test system for broader mutation spectrum analysis using the BC library with random oligonucleotide sequences; however, the assay using the *supF* gene presented in this paper has a significant advantage in both efficiency as a mutagenesis assay and cost-effectiveness. We believe that the method presented here will be used in further experiments by us and by other laboratories to advance the research on environmental mutagenesis, carcinogenesis, DNA repair, and will hopefully contribute to mutagenesis research by promoting accumulation of data for machine learning.

## Materials and methods
### Plasmids and plasmid constructions

The eight series of *supF* shuttle vector plasmids – pNGS2-K1, -K2, -K3, -K4, -A1, -A2, -A3, -A4, shown in *Figure 1—figure supplement 1*, were constructed from a pSB189L *supF* shuttle vector (*Parris and Seidman, 1992*; *Parris et al., 1994*; *Suzuki et al., 2022*) using NEBuilder HiFi DNA Assembly Master Mix (New England Biolabs, Ipswich, MA). Briefly, DNA fragments containing the amber suppressor tRNA gene (*supF*), the TP53-/Rb-binding-deficient mutant SV40 large T antigen (E107K/D402E) with the SV40 replication origin, the pBR327 origin of replication, the M13 intergenic region, or the ampicillin resistance gene (*amp'*) were amplified with the overlap sequences from pSB189L by PCR using high-fidelity KOD One PCR Master Mix (TOYOBO, Japan). DNA fragments containing the kanamycin resistance gene (*km'*) were amplified from an MCS1-EF1α-Blasticidin-pA-MCS2-PGK-hsvTK HR targeting vector (System Biosciences, Palo Alto, CA). The overlapping oligonucleotide primers were designed by NEBuilder Assembly Tool (New England Biolabs) and were synthesized commercially and purified on a reversed-phase column by FASMAC (Japan). The amplified DNA fragments were assembled to construct the plasmid DNAs by following the manufacturer's instructions (NEBuilder HiFi DNA Assembly Master Mix) and were transformed into *E. coli* HST08 competent cells (TaKaRa Bio Inc, Japan) for cloning. The sequences of the series of pNGS2 plasmid DNAs were confirmed by Sanger

sequencing. The plasmid DNAs were then transformed into *E. coli* JM109 competent cells (TaKaRa Bio Inc) for amplification, after which they were extracted using either GenElute Plasmid Miniprep or Midiprep plasmid DNA purification kit (Merck, Germany).

## Construction of the $N_{12}$-barcoded *supF* shuttle vector library

The pNGS2 plasmids (-A1-A4 or K1-K4) were digested at a unique site by incubation with *Eco*RV-HF restriction enzyme (New England Biolabs) at 37°C for 1 hr. The linearized pNGS2 was purified by agarose gel electrophoresis, followed by extraction using FastGene Gel/PCR Extraction kit (NIPPON Genetics, Japan). A quantity of 12 fmol purified *Eco*RV-digested pNGS2 was combined with 60 fmol of single-stranded oligonucleotides in a volume of 5 µL. The oligonucleotides containing randomized $N_{12}$ (12-nt sequences, 5'-dGGCCTCAGCGAATTGCAAGCTTCTAGAAGGCGATNNNNNNNNNNNNNAT CGAATTCGGATCCTTTCTCAACGTAA-3', where N=A, T, G, or C) were synthesized and purified on a reversed-phase column by FASMAC (Japan). The assembly reaction was performed using NEBuilder HiFi DNA Assembly Master Mix (New England Biolabs) following the manufacturer's instructions. The *E. coli* JM109 electro-competent cells were transformed with 1–2 µL of the assembly reaction mixture by electroporation. Following a 1 hr recovery in SOC medium at 37°C with agitation, the transformants were serially diluted, seeded on LB agar plates containing 25 µg/mL kanamycin, and incubated at 37°C overnight. The grown colonies were counted and harvested by scraping. Approximately $10^2$, $10^3$, and $10^4$ colonies were harvested for each desired complexity of $N_{12}$-BC sequences (pNGS2-K1-$N_{12}$-BC library: Lib$10^2$_#1, Lib$10^2$_#2, Lib$10^3$, and Lib$10^4$, pNGS2-K3-$N_{12}$-BC library: Lib$10^4$_#1 and Lib$10^4$_#2), and pNGS2-$N_{12}$-BC libraries were extracted from ~1.5 g of harvested *E. coli* JM109 cell pellets by using NucleoBond Xtra Midi (TaKaRa Bio Inc) and protocol for low copy plasmids. Each prepared library was interrogated for the presence of potential *supF* mutants by using the indicator *E. coli* strain RF01, and a mutation frequency of less than $10^{-4}$ was considered mutant free.

## Preparation of PCR templates for multiplexed NGS libraries

Each pNGS2-$N_{12}$-BC library was brought to a concentration of 100 ng/µL in sterile distilled water and exposed to UV-C radiation from germicidal lamps at a dose rate of 1.0 J/$m^2$·s for 0, 50, 100, and 200 s. For experiments in non-SOS induced *E. coli*, immediately after irradiation the $N_{12}$-BC library (5, 25, 50, and 100 ng for 0, 50, 100, and 200 J/$m^2$, respectively) was electroporated into the indicator *E. coli* strain RF01 and seeded on either titer LB agar plates containing 25 µg/mL kanamycin and 30 µg/ mL chloramphenicol, or selection plates containing 25 µg/mL kanamycin, 30 µg/mL chloramphenicol, 50 µg/mL nalidixic acid, and 100 µg/mL streptomycin. For experiments in mammalian cells, the UV-irradiated pNGS2-$N_{12}$-BC library was transfected into the human osteosarcoma U2OS cell line (HTB-96, American Type Culture Collection, Manassas, VA), with Lipofectamine 2000 (Thermo Fisher Scientific, Waltham, MA), as we have previously described (*Suzuki et al., 2022*). Briefly, U2OS cells were seeded into six-well plates at 3×$10^5$ cells/well the day before transfection, and transfection was carried out with 400 ng of pNGS2-$N_{12}$-BC library per well following the manufacturer's instructions. After 48 hr of culture, the transfected pNGS2-$N_{12}$-BC library was extracted from the cells (*Stary and Sarasin, 1992*). To digest unreplicated plasmids in the library, the library was treated with the restriction enzyme *Dpn* I which possesses specificity for methylated 5'-G$^m$ATC-3' sequences. The library was then transformed into RF01 following the already described transformation procedure. For both experiments – in *E. coli* and in mammalian cells, the *supF* mutant frequencies were estimated based on the number of colonies grown on the individual plates. The presumed number of required colonies was harvested as a cell pellet by scraping, and libraries were extracted using GenElute plasmid miniprep kit (Merck).

## Preparation of samples with unique indexes for multiplexed NGS

Each sample with a unique index for multiplexed NGS was prepared by PCR using KOD One PCR Master Mix (TOYOBO). For the standard protocol, 10 ng of the library (or from 1 to 100 ng for experiments presented in *Figures 2–4*) extracted from *E. coli* as described above was used for a 50 µL PCR reaction (30 cycles of amplification at 98°C for 10 s, 60°C for 5 s, and 68°C for 1 s) with a series of primer sets (*Figure 1—source data 1*) containing a pre-designed 6-nt sequence ($N_6$) as an index sequence for distinguishing each sample by sequence data analysis. PCR products were purified using NucleoSpin Gel and PCR Clean-up (TaKaRa Bio Inc), and then analyzed and quantified by Nano-drop spectrometer (Thermo Fisher Scientific) and agarose gel electrophoresis. The uniquely indexed

samples were combined together in the proper proportion according to the expected number of NGS reads for each sample. More than 100 reads per $N_{12}$-BC sequence were obtained for all samples in this study (approximately 400 and 16 Mb of sequencing data for the titer plates and the selection plates, respectively).

## DNBSeq sequencing and sequencing processing

Preparation of NGS libraries and sequencing were conducted by Seibutsu Giken Inc (Japan). The libraries were subjected to multiplexed deep sequencing in 200 bp paired-end mode on the NGS using a BGISEQ-G400 platform (MGI Tech, China). The concentration of the NGS samples was measured using either Synergy LX (Agilent Technologies, Santa Clara, CA) and QuantiFluor dsDNA System (Promega, Madison, WI), or Qubit 3.0 Fluorometer (Thermo Fisher Scientific) and dsDNA HS Assay Kit (Thermo Fisher Scientific). The libraries for DNBSeq were prepared using either MGIEasy FS DNA Library Prep Set (MGI Tech) (25 ng of sample, 8 cycles) or MGIEasy PCR-Free DNA Library Prep Set (MGI Tech) (200 ng of sample, enzymatic digestion for 4 min). The quality of the prepared libraries was assessed using an Agilent 2100 bioanalyzer and a High Sensitivity DNA kit (Agilent Technologies), or Fragment Analyzer and dsDNA 915 Reagent Kit (Agilent Technologies). The DNA nanoball was generated from the libraries using the MGIEasy Circularization Kit (MGI Tech).

## NGS data analysis

The FASTQ files generated from each $N_6$-index sample (sorting was performed by Seibutsu Giken Inc) were used for the following analysis: (1) The reverse-complement sequences were obtained from FASTQ data using SeqKit tools (*Shen et al., 2016*). (2) The paired-end reads from FASTQ data including the reverse-complement sequences were merged with a minimum 30 bp overlap region using CASPER (*Kwon et al., 2014*). (3) The merged sequences were mapped to a reference sequence, and only aligned sequences were converted to BAM files by using minimap2 (*Li, 2018*). (4) BAM files were converted to FASTQ files using SAMtools (*Li et al., 2009*; *Li, 2011*). (5) The extraction of the most likely true $N_{12}$-BC sequences with error corrections was performed based on a whitelist by UMI tools (*Smith et al., 2017*). Two 6-nt sequences, GGCGAT and ATCGAA, were used for the $N_{12}$-BC extraction, with only one base differences between $N_{12}$-BC sequences being allowed, and corrections were implemented using the following regular expression: (".*(GGCGAT) {s ≤ 0}(?P<cell_1>.{12})(?P<umi_1>.{0})(ATCGAA){s ≤ 0}.*"). (6) Using VarDict, the sequence data for all $N_{12}$-BC sequences with more than 9 reads per $N_{12}$-BC was employed for the alignment to the reference sequence, variant calling, and calculation of allele frequencies (*Lai et al., 2016*). (7) Spotfire version 7.11.1 (TIBCO, Palo Alto, CA) was used for data organization, filtering, and visualization.

## Acknowledgements

A part of this study was carried out at the Analysis Center of Life Science, Natural Science Center for Basic Research and Development, and the Joint Usage/Research Center Program of the Research Institute for Radiation Biology and Medicine (RIRBM), Hiroshima University. We would like to thank Dr. Elena K Zaharieva for her invaluable contribution to the English editing.

## Additional information

### Funding

| Funder | Grant reference number | Author |
| --- | --- | --- |
| Japan Society for the Promotion of Science | JP 19K123222 and JP 22K12375 | Hidehiko Kawai |

The funders had no role in study design, data collection and interpretation, or the decision to submit the work for publication.

## Author contributions
Hidehiko Kawai, Conceptualization, Resources, Data curation, Formal analysis, Supervision, Funding acquisition, Validation, Investigation, Visualization, Methodology, Writing – original draft, Project administration, Writing – review and editing; Ren Iwata, Investigation; Shungo Ebi, Ryusei Sugihara, Investigation, Methodology; Shogo Masuda, Resources, Investigation; Chiho Fujiwara, Data curation, Formal analysis, Investigation; Shingo Kimura, Formal analysis; Hiroyuki Kamiya, Conceptualization, Supervision, Funding acquisition, Writing – review and editing

## Author ORCIDs
Hidehiko Kawai ⓘ http://orcid.org/0000-0003-2213-7166
Hiroyuki Kamiya ⓘ http://orcid.org/0000-0001-6866-5322

## Decision letter and Author response
Decision letter https://doi.org/10.7554/eLife.83780.sa1
Author response https://doi.org/10.7554/eLife.83780.sa2

## Additional files

### Supplementary files
• Supplementary file 1. The valid sequences and the numbers of N12-BCs in the library.

• Supplementary file 2. The data of variant calling analysis by VarDict for the N12-BC libraries used for *Figures 2 and 3*.

• Supplementary file 3. The data of variant calling analysis by VarDict for the N12-BC libraries used for *Figures 4 and 5*.

• Supplementary file 4. The data of variant calling analysis by VarDict for the N12-BC libraries used for *Figures 6 and 7*.

• Supplementary file 5. The data of variant calling analysis by VarDict for the N12-BC libraries used for *Figures 8 and 9*.

• Supplementary file 6. The data of variant calling analysis by VarDict for the N12-BC libraries used for *Figure 10*.

• Supplementary file 7. The data of variant calling analysis by VarDict for the N12-BC libraries used for *Figure 12*.

• MDAR checklist

### Data availability
NGS raw data has been uploaded to NCBI under the accession number PRJDB13753 (DRA014420 for Figures 2 and 3, DRA014421 for Figures 4-7, DRA014422 for Figures 8-11, DRA014422 for the samples without transformation into *E. coli*).

The following dataset was generated:

| Author(s) | Year | Dataset title | Dataset URL | Database and Identifier |
|---|---|---|---|---|
| Kawai H | 2022 | Advanced SupF mutagenesis assay | https://www.ncbi.nlm.nih.gov/bioproject/PRJDB13753 | NCBI BioProject, PRJDB13753 |

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
