## [Editor Report]

This is a comprehensive description of a technical advance for the analysis of mutations based on the most widely used system for reporting mutations in mammalian, including human, cells. As costs for NGS decline it is likely to become the approach of choice.

---

## [Decision Letter]

[Editors' note: this paper was reviewed by Review Commons.]

---

## [Author Response]

Reviewer #1 (Evidence, reproducibility and clarity (Required)):The analysis of mutations in mammalian, including human, genomes has been of interest for many decades. Early DNA sequencing technologies enabled direct identification of mutations in target genes provided that the mutant genes could be readily isolated. This requirement stimulated the development of shuttle vector plasmids that carried a mutation marker gene and could replicate in both mammalian and bacterial cells. These were used in experiments in which the plasmids, treated with a mutagen, would be passaged through mammalian host cells after which the progeny plasmids were introduced into an indicator bacterial strain. Colonies with mutant marker genes could be distinguished by color or survival, the plasmids recovered, and the sequence of the mutant gene determined. The shuttle vector plasmid that became the most widely used contained as the marker the supF amber suppressor tyrosyl tRNA gene positioned in the plasmid such that deletion mutations associated with mammalian cell transfection were selected against. Although various improvements have been introduced since its introduction in the mid-1980s, including bar codes to distinguish independent from sibling mutations (in the early 1990s), the basics of the system have been maintained, and it and variations are still in use.The Kamiya group has made several adjustments to the supF shuttle vectors, including the construction of indicator bacterial strains based on survival of bacteria containing mutant supF genes (the initial system relied on colony color). They have published many studies of mutagenesis by various agents, error prone polymerases, etc. In the current submission they describe a comprehensive approach to identifying mutations in the supF gene that exploits Next Generation Sequencing technology that can identify the full spectrum of mutations including those that escape detection in phenotypic screens. The study is exhaustive and presents a methodical validation of each component of their approach. They report UV induced mutations, the mechanism of which has been well characterized in previous literature. They also describe a category of multiple mutations, which had been observed in the early work with the supF plasmids, and whose relationship to UV photoproducts is most likely indirect.

We thank the Reviewer for their very insightful feedback to our manuscript and their positive assessment. We have added some discussion points based on the essential references mentioned in the Reviewer’s comments, which we believe made the explanation of our study more complete.

Major comments:This manuscript presents a technical advance on the use of the supF mutation reporter system. The extent of the validation of each component of the system, including the bar code is rigorous. Their data on the nature and location of UV induced mutations are in very good agreement with previous studies with supF and other reporter genes, a further validation of their approach. Their discussion of the mechanism of the UV induced mutations is in accord with prior work from other laboratories. However, their interpretation of the multiple mutations, although reasonable in invoking a role for APOBEC deamination of cytosines (see eLife. 2014; 3: e02001 for another discussion of this issue), overlooks a much earlier study on the same topic that showed that nicks in the vicinity of the marker gene are mutagenic and can induce multiple mutations (Proc Natl Acad Sci 1987 84:4944-8). It would be useful for the authors to consider their data on the multiple mutations in the light of the earlier analysis. Furthermore, a check to verify the covalently closed circular integrity of the plasmid preparations would be an important quality control and could reduce the mutagenesis observed in 0 UV controls.

We thank the Reviewer for the valuable comments that made our manuscript clearer and more emphatic. We are hereby addressing all of the Reviewer’s concerns. The available data accumulated from previous studies have proved the high sensitivity of the supF assay as a mutagenesis assay, which now has been clearly supported by the results in the current study. We believe that this NGS assay will be able to fulfil the data requirements to tackle many questions related to mutagenesis, thanks to the simplicity and cost-effectiveness of the procedure. However, to meet the experimental objectives, the preparation and analysis of the library are crucially important procedures in the stages of initial setting up of the assay. The covalently closed circular integrity of the vector library is definitely one of the important points we should pay attention to when performing this assay. After the construction of the BC_12_-library, we have to check the quality of the library by agarose gel electrophoresis. The background mutation frequency and the sequence of the library itself (uploaded as described in the data availability section of this manuscript) also needs to be analyzed by NGS before the experiment. We are also routinely constructing the double-stranded shuttle vector from a single-stranded circular DNA with a variety of site-specific damaged oligonucleotides. The treatment with T5 exonuclease followed by purification is absolutely essential to decrease the background mutation frequency. Without the treatment with the exonuclease, cluster mutations may be increased under specific experimental conditions. For this study, we carried out the conventional supF assay using the BC_12_-library purified after T5 exonuclease treatment. However, in this case the process of purification slightly increased the mutant frequency of the BC_12_-library to about 2 x10^-4^ (corresponding to 1x10^-6^/bp).Therefore, when setting up the essay, we have to consider the background control that we will need for the data analysis. In response to the Reviewer’s comments, we have now added the following paragraph in the Discussion section:

Page 16, line 25:

”(5) For the supF assay, spontaneous cluster mutations at TC:GA sites were often observed, and it was well illustrated in an earlier study that a nick in the shuttle vector was a trigger for these asymmetric cluster mutations (54). Therefore, we need to be aware of the quality of each library and how it affects the outcome of each analysis, especially for detection of very low levels of mutations. Depending on the purpose of the experiments, in the preparation of covalently closed circular vector libraries it is essential to eliminate the background level of mutations. In fact, the in vitro construction of the library of double-stranded shuttle vectors from single-stranded circular DNA requires the process of treatment with T5 Exonuclease, which drastically decreases background mutations.”

Minor pointsThe authors state that only 30% of the base sequence of the supF gene can be "used for dual-antibiotic selection on the indicator *E. coli*". An earlier review (Mutation Res 220: 61,1989) indicated that within the mature tRNA region single or tandem mutations had been reported at 87% of sites, using the colony color assay. The direct NGS analyses would be indifferent to phenotype, and one would expect the maximum number of mutable sites would be recovered from this approach. It would be helpful for an explicit statement regarding the number of mutant sites to be in the Discussion, as this should strengthen the case for the NGS strategy.

We thank the Reviewer for the helpful comment. These are important points we should indeed mention. This method will complement previous data, and especially the data from titer plates will provide us with non-biased mutation spectra for the whole analyzed region. We have now explained in detail about the coverage of mutation spectra in the discussion section.

Page 14, line 14:

“The mutation spectra of single or tandem base-substitutions for inactive supF genes identified by using the blue-white colony color assays were comprehensively summarized in an earlier review article, and it was noted that the mutations were detected at 86 sites within a 158-bp region covering the supF gene (54%) and at 74 sites within the 85-bp mature tRNA region (87%), thus demonstrating the great sensitivity of the supF assay system for analysis of mutation spectra (19). However, obtaining reliable datasets by the conventional supF assay requires skill and experience, especially for studies where the mutations of interest are induced with low frequency. The method has been advanced by the construction of indicator bacterial strains with different supF reporter genes which allow selection based on survival of bacteria containing mutant supF genes. However, the fact that the supF phenotypic selection process relies on the structure and function of transfer RNAs that may be differently affected by different mutations means that the improvement of the efficiency of the selection process may cause loss of coverage of the mutation spectra, as it is under our experimental conditions, where the coverage is about 30% (19,20).”

Page 15, line 4:

“From this point of view, we believe that we can secure a sufficient number of experiments to improve the accuracy of the analysis and to confirm the reproducibility of the experiments. Furthermore, the data from colonies grown on titer plates provides us, at least in principle, and with the exception of large deletions and insertions, with non-biased mutation spectra for the whole analyzed region.

Supplementary Figure 1 shows the organization of 8 supF reporter plasmids. Were these discussed in the text and employed in the experiments? It was not clear in the text.

We thank the Reviewer for the helpful comment. It was indeed not clear which vectors we used and why we constructed a series of vectors. Now, we have added the vectors we used for the constructions of the library and each experiment in the RESULTS and MATERIALS AND METHODS sections. Since this is quite important for us and, we believe, the readers, we also added the explanations in the Discussion section, detailing why we have constructed a series of shuttle vectors, as follows:

Page 19, line 36:

“Mutational signatures identified in cancer cells are emerging as valuable markers for cancer diagnosis and therapeutics. Innumerable physical, chemical and biological mutagens, including anticancer drugs, induce characteristic mutations in genomic DNA via specific mutagenic processes. The mutation spectra obtained here by using the presented advanced method were in good agreement with accumulated data from previous papers where the conventional method had been used, with the advantage that our method provided less-biased mutation spectra data. As described above, the datasets presented here highlighted novel mutational signatures and also cluster mutations with a strand-bias, which could be associated with the processes of replication, transcription, or repair of DNA-damage, including a single strand break (a nick). In this study, eight series of supF shuttle vector plasmids were constructed, as presented in Supplementary Figure S1; however, the analysis was carried out using N_12_-BC libraries prepared from either pNGS2-K1 (Figures 1-4) or pNGS2-K3 (Figures 5-10). The pNGS2-K1/-A1/-K4/-A4 and pNGS2-K2/-A2/-K3/-A3 vector series contain an M13 intergenic region with opposite orientations relative to the supF gene, which allows us to incorporate specific types of DNA-damage at specific sites in the opposite strand of the vector library. Also, the pNGS2-K1/-A1/-K3/-A3 and pNGS2-K2/-A2/-K4/-A4 vector series contain the SV40 replication origin, which enables bidirectional replication and transcription, at opposite sides of the supF gene. Although this is still preliminary data, it is notable that the spontaneously induced mutations for the different vectors in U2OS cells were not significantly different. Therefore, the here presented mutagenesis assay with NGS, by using these series of libraries, can be applied in many different types of experiments to address both quantitative and qualitative features of mutagenesis. It is possible to design series of libraries containing DNA lesions or sequences suitable for the investigation of specific molecular mechanisms, such as TLS, template switching, and asymmetric cluster mutations.”

Cross-consultation commentsComment on the issue raised by Reviewer #2 regarding plasmids with unrepaired DNA damage introduced into *E. coli* after passage through U2OS cells: treatment of the plasmid harvest with Dpn1 eliminates un-replicated plasmid DNA. Also, SV40 T antigen drives run away replication of the plasmids, which contain the SV40 origin of replication. This greatly dilutes plasmids with remaining UV photoproducts.Reviewer #1 (Significance (Required)):SignificanceThis is a comprehensive description of a technical advance for the analysis of mutations based on the most widely used system for reporting mutations in mammalian, including human, cells. As costs for NGS decline it is likely to become the approach of choice.Reviewer #2 (Evidence, reproducibility and clarity (Required)):In this manuscript, the authors developed a novel mutagenesis assay by combining the conventional supF forward mutagenesis assay with NGS technology. The manuscript is well written, providing design, methods, and results of the experimental system in very much details, which this reviewer highly evaluates. However, the manuscript may be too long and could be more concise. In addition, this reviewer is afraid that main figures seem difficult to fit printed pages (especially multi-paneled figures of large size, such as Figure 5 through 8). The authors should re-organize the figures by reducing size and/or moving partly to supplementary information.

We thank the Reviewer for the helpful comments to our manuscript. It is true that the multi-paneled figures were too large, and we have now re-analyzed and optimized most of the figures by reducing size, transferring to Supplementary Figures, and separating one figure into two. Although the number of Figures and Supplementary Figures have now increased, we believe that it has become easy to follow for readers and to fit printed pages. We considered carefully the Reviewer’s remark about the length of the manuscript, but we feel that the text was already as concise as we could make it, and we have already left out some more detailed explanations.

Specific comments1. Some UV-induced DNA damage (typically CPD) is repaired only slowly in human cells, so that the replicated plasmid DNAs recovered from U2OS cells may still contain damage and possibly induce mutations in *E. coli* after transfection. As the result of high sensitivity of NGS analysis, it is worried that such mutations could be also included in the results. To obtain even more accurate mutational characteristics in mammalian cells, the authors could consider to treat the DNA samples with photolyases before transformation of E. coli. The authors could consider to discuss on this point.

We thank the Reviewer for the helpful comment, indeed Dpn I treatment is one of the very important procedures for avoiding analysis bias. We have now expanded the explanation why the libraries have to be treated with Dpn I, as follows:

Page 11, line 4:

“the libraries were extracted from the cells, and treated with dam-G^m^ATC-methylated DNA specific restriction enzyme Dpn I to digest un-replicated DNAs that contain UV-photoproducts.”

2. It is quite intriguing that multiple mutations in a single BC clone tend to occur in the same DNA strand. Is there any trend in a distance between the mutated sites? Considering participation of TLS polymerases in the first round of replication, it may be interesting if multiple DNA lesions occur in relatively close positions so that TLS polymerases elongate the DNA strand without switching back to replicative polymerases.

We thank the Reviewer for the valuable and insightful suggestions for this assay. We have analyzed the positions of SNSs in multiple-mutations shown in Supplementary Figures S11 and S12. As the reviewer mentioned, we may be able to address the mechanisms of TLS switching in mammalian cells by using this assay. In this study, the obtained non-biased mutation spectra of multiple mutations may not be enough for the static analysis, but our results indicate that multiple mutations were induced at relatively close positions. It would be interesting if we could address the mechanisms of TLS polymerase switching. We believe that the accumulation of large numbers of non-biased mutation spectra will provide us with growing opportunities to address more questions in mutagenesis. We have now added the Supplementary Figures S11 and S12, as well as the following discussion points:

Page 14, line 6:

“(5) The distance between two SNSs in multiple mutations induced by UV irradiation was relatively shorter than the theoretically expected based on the sequence (Supplementary Figures S11 and S12).”

Page 18, line 27:

“In addition, the positions of SNSs in the multiple mutations were closer to each other compared to the theoretically expected positions (Supplementary Figures S11 and S12), which may reflect switching events involving TLS polymerases. It should be noted that the presented data for the distance between two SNSs in the multiple mutations was analyzed from the data from selection plates in order to secure a sufficient number of mutations, and therefore, there may be a bias due to hot spots associated with the selection process. However, the results from the limited number of mutations from the titer plates are similar to these from the selection plates. It can be proposed that this assay may also be applied for analysis of TLS polymerases in mammalian cells.”

3. This reviewer is wondering whether the results of mammalian cells are influenced by transcription-coupled repair in this experimental system. Because the SV40 replication origin functions as bidirectional promoters, the supF region may be transcribed in U2OS cells so that DNA damage on transcribed strands may be removed more efficiently than non-transcribed strands. Please comment on this, if relevant.

We thank the Reviewer for the insightful comments. This issue is also very important and interesting, and should be addressed in the mutagenesis research. That is exactly the reason why we presented series of vectors for the assay in this paper. The SV40 replication origin has an effect on the background mutations, which this is also dependent on the experimental conditions. However, this needs to be confirmed by further studies. We hope the idea for these constructions will be helpful for many laboratories. We have now added the following parts in the Discussion section.

Page 18, line 36:

“Mutational signatures identified in cancer cells are emerging as valuable markers for cancer diagnosis and therapeutics. Innumerable physical, chemical and biological mutagens, including anticancer drugs, induce characteristic mutations in genomic DNA via specific mutagenic processes. The mutation spectra obtained here by using the presented advanced method were in good agreement with accumulated data from previous papers where the conventional method had been used, with the advantage that our method provided less-biased mutation spectra data. As described above, the datasets presented here highlighted novel mutational signatures and also cluster mutations with a strand-bias, which could be associated with the processes of replication, transcription, or repair of DNA-damage, including a single strand break (a nick). In this study, eight series of supF shuttle vector plasmids were constructed, as presented in Supplementary Figure S1; however, the analysis was carried out using N_12_-BC libraries prepared from either pNGS2-K1 (Figures 1-4) or pNGS2-K3 (Figures 5-10). The pNGS2-K1/-A1/-K4/-A4 and pNGS2-K2/-A2/-K3/-A3 vector series contain an M13 intergenic region with opposite orientations relative to the supF gene, which allows us to incorporate specific types of DNA-damage at specific sites in the opposite strand of the vector library. Also, the pNGS2-K1/-A1/-K3/-A3 and pNGS2-K2/-A2/-K4/-A4 vector series contain the SV40 replication origin, which enables bidirectional replication and transcription, at opposite sides of the supF gene. Although this is still preliminary data, it is notable that the spontaneously induced mutations for the different vectors in U2OS cells were not significantly different. Therefore, the here presented mutagenesis assay with NGS, by using these series of libraries, can be applied in many different types of experiments to address both quantitative and qualitative features of mutagenesis. It is possible to design series of libraries containing DNA lesions or sequences suitable for the investigation of specific molecular mechanisms, such as TLS, template switching, and asymmetric cluster mutations.”

4. page 13: Please check whether the description of Figure 9C is correct (6th line, graph on top; 9th line, bottom graph).

We thank the Reviewer for carefully checking our manuscript, it was mislabeled in the text. Now, following the Reviewer’s comments, most figures have been changed from the figures in the previous submission. We appreciate the careful review.

Cross-consultation commentsReviewer #1 gives quite relevant comments as an expert of the mutagenesis field. It would improve this manuscript greatly for the authors to make appropriate modifications according to his/her suggestions.Reviewer #2 (Significance (Required)):It is quite convincing that this method has a great potential to give much more extensive information on mutational characteristics, most importantly, by eliminating the bias caused by phenotypic selection. Therefore, this work certainly must be worth being published in an appropriate journal.